

# A high-resolution nested model to study the effects of alkalinity additions in Halifax Harbour, a mid-latitude coastal fjord

Arnaud Laurent[1], Bin Wang[1], Dariia Atamanchuk[1], Subhadeep Rakshit[1†], Kumiko Azetsu-Scott[2], Chris Algar[1], and Katja Fennel[1]

[1] Department of Oceanography, Dalhousie University, Halifax, Nova Scotia, Canada
[2] Bedford Institute of Oceanography, Dartmouth, Nova Scotia, Canada
[†] Present address: Program in Atmospheric and Oceanic Sciences, Princeton University, Princeton, NJ, USA

*Correspondence to*: Arnaud Laurent (arnaud.laurent@dal.ca)

**Abstract.** Surface ocean alkalinity enhancement (OAE), through the release of alkaline materials, is an emerging marine carbon dioxide removal technology that could increase the storage of anthropogenic carbon in the ocean. Observations collected during recent and on-going field trials will provide important information on the feasibility and effects of alkalinity additions on carbon cycling and study ecological responses. However, given the scales involved (24/7 continuous addition, meters to hundreds/thousands of kilometers and minutes to months for alkalinity dispersion) observations, even with the use of autonomous platforms, will remain inherently sparse and limited. Alone, they cannot provide a comprehensive quantification of the effects of OAE on the carbonate system, and ultimately of the net air-sea $CO_2$ fluxes. Numerical models, informed and validated by field observations, are therefore essential to OAE deployments and the measurement, reporting, and verification (MRV) of any resulting carbon uptake. They can help guide fieldwork design, including optimal design of measurement monitoring networks, provide forecasts of the ocean state, simulate the effects of alkalinity additions on the seawater carbonate system, and allow one to quantify net $CO_2$ uptake. Here we describe a coupled physical-biogeochemical model that is specifically designed for coastal OAE. The model is an implementation of the Regional Ocean Modelling System (ROMS) in a nested grid configuration with increasing spatial resolution from the Scotian Shelf to Halifax Harbour (coastal fjord, eastern Canada), a current test site for operational alkalinity addition. The biogeochemical model simulates oxygen dynamics, carbonate system processes (including air-sea gas exchange), and feedstock properties (dissolution, sinking). We present a multi-year hindcast validated against the long-term weekly time series available for a long-term monitoring station at the deepest part of Halifax Harbour, as well as alkalinity addition simulations at various locations inside and outside the harbour to show the model's capabilities for assessing the effects of OAE at this coastal site.

## 1 Introduction

Global temperature has increased by more than 1.1°C from pre-industrial average and is likely to exceed, at least temporarily, the +1.5°C goal of the Paris agreement in the coming decade (WMO, 2024a, b). At the current rate of greenhouse gas emissions, warming is expected to reach +2.7°C by the end of the century (UNEP, 2023). Without rapid and significant mitigation actions, changes in the climate system (increasingly irreversible) will severely affect





ecosystems, biodiversity, as well as the livelihood of current and future generations (IPCC, 2023). As the urgency increases, carbon dioxide removal (CDR) techniques have been proposed as a supplement to emission reductions.

CDR can accelerate progress toward net zero $CO_2$ emission by compensating for hard-to-abate residual emissions (e.g. from agriculture, aviation, shipping or industrial sectors) and remove legacy anthropogenic $CO_2$ from the atmosphere (IPCC, 2023). It is estimated that 10 Gt of $CO_2$ residual emissions need to be compensated annually by 2050 and 20 Gt by 2100 to achieve net zero $CO_2$ emissions by the end of the century (NASEM, 2019).

Current CDR measures involve land-based techniques such as afforestation, reforestation or carbon capture

and storage (Nabuurs et al., 2023; Riahi et al., 2023). The risks, such as competition for land, non-permanence, energy cost, depend on the method and the scale involved (Fuhrman et al., 2020). Given the large reservoir of carbon in the ocean and its potential for additional storage, marine CDR methods (mCDR) have gained interest in the last decade (Gattuso et al., 2021; Oschlies et al., 2025). Seaweed cultivation, nutrient fertilization (e.g., iron) or artificial upwelling have been proposed to increase the ocean's carbon storage (NASEM, 2022). However, the risks associated to these

biotic mCDR technologies may be important because they modify the biological carbon pump and nutrient cycles, with potential important side effects on marine ecosystems (Cullen and Boyd, 2008; Fennel, 2008; Oschlies et al., 2010; Wu et al., 2023). The impact on the deep ocean is of particular concern (Levin et al., 2023). The effectiveness of biotic mCDR technologies is therefore uncertain (Oschlies et al., 2025). Ocean alkalinity enhancement (OAE) is an abiotic method that aims to store additional inorganic carbon in the ocean via the addition of alkalinity at the

surface. In theory, this technique has fewer risks to marine ecosystems because it does not modify biological cycles, although the side effects depend on the source material and technology used and possibly on the addition level (Delacroix et al., 2024; Ferderer et al., 2024; Guo et al., 2024). If scaled up to the global ocean, OAE could remove > 1 Gt $CO_2$ yr$^{-1}$ from the atmosphere (Renforth and Henderson, 2017).

The challenge for all mCDR methods is to detect and monitor their effects on the carbon pool, and to quantify

the carbon removed from the atmosphere (Fennel, 2025). For OAE the effects of the added alkalinity on the solubility pump and the environment need to be quantified from tens of meters near the release site to possibly thousands of kilometers until addition waters are equilibrated with the atmosphere. Moreover, this signal needs to be detected on top of the natural variability in carbonate system parameters, which is challenging especially in coastal settings (Carter et al., 2019). Because the perturbations due to OAE are overlayed on top of natural variability of carbonate system

parameters, which is controlled by a combination of biotic and abiotic factors, observations alone are insufficient in characterizing the effects of OAE (Bach et al., 2023).

To quantify how much carbon is removed, an essential aspect of measurement, reporting, and verification (MRV), the addition needs to be monitored and compared with a counterfactual case without addition. By definition, it is not possible to measure the counterfactual (Fennel, 2025). Therefore, fit-for-purpose validated models are

essential for quantifying any net carbon uptake resulting from alkalinity addition (Fennel et al., 2023; Ho et al., 2023).

Global ocean models represent the large-scale circulation and therefore have been used to assess the feasibility of OAE and track its effects at the global scale (Burt et al., 2021; Feng et al., 2017; Ilyina et al., 2013; Köhler et al., 2013; Lenton et al., 2018; Palmiéri and Yool, 2024; Tyka, 2025). The limitation of these models is their coarse resolution which does not allow them to represent mesoscale processes and poorly resolves dynamics on coastal



and shelf scales (Laurent et al., 2021). For practical reasons, e.g., pre-existing infrastructure such as sewage and cooling outfalls, and to reduce the carbon footprint of alkalinity delivery, nearshore areas will likely be major sites of alkalinity addition in OAE projects (Eisaman et al., 2023). High resolution regional models are better suited to simulate alkalinity addition at these scales (10 m–100 km) than global models. The addition signal can then be passed on to a global model for basin scale tracking of unequilibrated alkalinity or to an earth system model to assess the long-term,
global re-equilibration of the mobile carbon pool (Fennel, 2025).

Wang et al. (2025) recently developed a regional model to study OAE in the Halifax Harbour on the eastern coast of Canada. Halifax Harbour has been in operation as an OAE test site since 2023 (Cornwall, 2023). Wang et al.'s (2025) study highlighted the importance of local processes, such as local circulation and residence time, and feedstock characteristics such as its dissolution and sinking rates on the fate of the added material. The likelihood of
ecosystem exposure to alkaline waters was also simulated and quantified by Wang et al.'s (2025). It is clear from Wang et al.'s (2025) results that added alkalinity doesn't simply disperse onto the shelf after addition, as assumed in global models, but is retained for some time inshore, depending on the dissolution and sinking rates of the alkaline material used and seasonal circulation features (i.e., residence time). The model version in Wang et al. (2025) did not include an explicit representation of the carbonate system and therefore was not used to assess addition effects on
carbonate system parameters nor to quantify the resulting local net $CO_2$ uptake. Other regional modelling studies including explicit carbonate system parameters (e.g., Khangaonkar et al., 2024; Li et al., 2025; Wang et al., 2023) have been carried out but not for this location. OAE studies usually use generic, previously validated biogeochemical models that include carbon chemistry but that were not explicitly tailored to OAE research. Most models assume the addition of fully dissolved rather than particulate material (e.g., Kwiatkowski et al., 2023; Nagwekar et al., 2024;
Palmiéri and Yool, 2024; Zhou et al., 2025), whereas mineral feedstock particles will dissolve over time (Schulz et al., 2023). Feedstock characteristics have a significant effect on the location and time scale of surface expression of the addition signal  (Wang et al., 2025). Recent attempts have been made, at the global scale, to adapt biogeochemical models to study OAE. Zhou et al. (2025) used two carbonate systems to simulate the addition and counterfactual cases, but without dissolution kinetics. Fakhraee et al. (2023) developed a biogeochemical-mineral dissolution model but
within a coarse physical framework.

To fill the gaps described above, we present a high-resolution, coupled physical-biogeochemical-addition-dissolution model that is designed to support OAE research in coastal environments. The model builds on Wang et al. (2025) to include oxygen and carbonate system chemistry in a realistic alkalinity addition setting. The model is validated and tested for Halifax Harbour. The observations used for the study are described in Section 2. The
circulation model is presented in Section 3 and the results of an eight-year simulation (2016-2023) validated against observations on the shelf and in the harbour are presented. The biogeochemical model is described in Section 4 and validated for 2016-2023. The addition model is then presented and tested with a series of sensitivity simulations in Section 5. The overall results are discussed in Section 6.

## 2    Supporting observations

### 2.1   Station 2 on the Scotian Shelf



Station 2, located on the Scotian Shelf in 150 m water depth (44° 16' 01" N, 63° 19' 01" W), is sampled biweekly or monthly by the Atlantic Zone Monitoring Program (AZMP). The observations include CTD casts (temperature, salinity, oxygen)(MEDS, 2024) and Niskin bottle samples at various depths for dissolved inorganic carbon (DIC) and alkalinity (Cyr et al., 2022).

**2.2 Compass Buoy station in Halifax Harbour**

The Bedford Basin Monitoring Program (BBMP) provides weekly observations, since 1992, at the Compass Buoy station in Halifax Harbour's Bedford Basin (the deep basin at the head of the harbour). This station is at the deepest location of the harbour, in the center of Bedford Basin (70 m depth, 44° 41' 37" N, 63° 38' 25" W)(Fisheries and Oceans Canada, 2025). The observations include CTD casts (temperature, salinity, oxygen) and Niskin bottle samples

(for various biological and chemical variables) at 4 depths (0, 5, 10 and 60 m). Carbonate system measurements (DIC, alkalinity) have been collected since 2016.

pCO$_2$ was calculated from TA and DIC with CO2SYS (pyCO2SYS, Humphreys et al., 2022) using the "Mehrbach refit" dissociation constants for carbonic acid (Dickson and Millero, 1987; Mehrbach et al., 1973) and the "Dickson" dissociation constants for bisulfate ions (Dickson, 1990). Bottle samples reported in units of µmol kg$^{-1}$ were converted

to mmol m$^{-3}$ assuming a conversion factor of 1.025 kg m$^{-3}$.

**2.3 Benthic fluxes and respiration**

Sediment-water fluxes, benthic and water column respiration were measured at several locations within the harbour using a benthic chamber and sediment core profiles (Rakshit et al., 2023). These data were used to parameterize the biogeochemical model (see Section 4.1).

**3 Circulation model and shelf biogeochemistry**

The model is a regional implementation of the Regional Ocean Modelling System (ROMS, version 3.9) for the Scotian Shelf and the Halifax Harbour (Figure 1). ROMS is a free-surface, primitive equation ocean model with terrain-following vertical coordinates (Haidvogel et al., 2008). The model has 40 vertical layers with increasing resolution near the surface and bottom; the horizontal resolution is different for each nested grid (see below). Advection is

prescribed with a third-order upwind (horizontal) and a fourth order centered (vertical) scheme for momentum equations and with the High-order Spatial Interpolation at the Middle Temporal level scheme (HSIMT; Wu, 2014) for biological tracers. Vertical mixing is parameterized using the nonlocal K-profile parameterization (Durski et al., 2004; Large and Gent, 1999).

**3.1 Nested model grids**

The model is setup with 3 nested grids of increasing resolution focussing in on Halifax Harbour (Figure 1c,d). The coarsest grid, ROMS-H1 (Figure 1a,c), covers the central Scotian Shelf with an average horizontal resolution of 760 m and a bathymetry interpolated from GEBCO (https://www.gebco.net). The grid is run with a biogeochemical model



that includes the nitrogen cycle (Fennel et al., 2006; Laurent et al., 2021), phosphorus cycling (Laurent et al., 2012), oxygen dynamics (Fennel et al., 2013), and carbonate chemistry (Fennel, 2010; Laurent et al., 2017). This model is referred subsequently as the explicit biogeochemical model. It is forced with tides and reanalysis data at the open boundaries (see Section 3.2 below). A similar implementation of the model was used for larger-scale implementations of the northwest North Atlantic shelf and adjacent open ocean and shown to represent physical and biogeochemical dynamics well (Ohashi et al., 2024; Rutherford et al., 2021). The ROMS-H2 domain is nested within ROMS-H1 with a refinement factor of 5 that covers the inner Scotian Shelf and the Halifax Harbour (Figure 1c). It has a horizontal resolution of 150 m and is forced at the open boundaries by interpolated results from ROMS-H1. The bathymetry is generated from the NONNA-10 and NONNA-100 products of the Canadian Hydrographic Services (https://open.canada.ca/data/en/dataset/d3881c4c-650d-4070-bf9b-1e00aabf0a1d), supplemented by GEBCO where high-resolution data from the NONNA products are unavailable. ROMS-H3 is the highest-resolution domain, nested within the ROMS-H2 grid with a refinement factor of 3 and covers Halifax Harbour (Figure 1b-d). The grid has a horizontal resolution of 50 m, and its bathymetry is generated from NONNA-10. ROMS-H3 can be run with two-way nesting or with offline one-way nesting. For the two-way nesting case (called ROMS-H23), ROMS-H2 and ROMS-H3 are run simultaneously and exchange information at every time step. For the offline one-way nesting case, ROMS-H3 open boundary conditions are calculated offline using results from ROMS-H2. Hereafter, most of the model validation and analysis use ROMS-H2. Simulations with ROMS-H3 are presented in Section 5.2.4. Results of ROMS-H23 are not presented here, but are available in Wang et al. (2025).

## 3.2 Forcing

The 3 ROMS domains share the same atmospheric forcing, prescribed hourly from the ERA5 reanalysis (Hersbach et al., 2023). Initial and open boundary conditions in ROMS-H1 are set by the GLORYS12V1 reanalysis product (https://doi.org/10.48670/moi-00021; Lellouche et al., 2021) for the physical variables and with a daily climatology calculated from the multiyear simulation of Rutherford et al. (2021) for the biogeochemical variables. ROMS-H1 is also forced by tides using 9 tidal constituents extracted from the ADCIRC Tidal Database version EC2001_v2e (https://adcirc.org/products/adcirc-tidal-databases/; Mukai et al., 2002) and has a surface heat flux correction towards the Multi-scale Ultra-high Resolution (MUR) sea surface temperature (SST) daily product (http://dx.doi.org/10.5067/GHGMR-4FJ04; Chin et al., 2017) with a relaxation time scale of 10 days.

River forcing is added using daily discharge observations from the National Water Data Archive (HYDAT) database (https://wateroffice.ec.gc.ca) for the LaHave River (01EF001; ROMS-H1) and Sackville River (01EJ001; all grids). The Halifax Harbour includes additional freshwater sources from five wastewater treatment plants (WWTP, Figure 1d) and 11 other minor inflows. Freshwater discharge at Mill Cove WWTP uses a daily climatology calculated with a multiannual daily discharge time series from Halifax Water. Other WWTP are scaled from Mill Cove using annual discharge information available from Halifax Water, following Wang et al. (2025). Minor inflows are scaled to the Sackville River using average discharge information from Buckley and Winters (1992), summarized in Table 1 of Shan (2010). River nutrients and carbonate data are set constant based on observations collected in the Sackville River (Sackville River + minor inflows) and at Mill Cove (McGinn et al., 2012; WWTPs).



Atmospheric $pCO_2$ is parameterized using a fit to the monitoring observations from Sable Island collected
by the Environment Canada Greenhouse Gas Measurement Program and available in the GLOBALVIEW database
(Schuldt et al., 2023):

$$pCO_{2,air} = a_0 + a_1 t + a_2 t^2 + 1.25 a_3 \times \log(\sin(2\pi t + a_4) + a_5), \qquad (1)$$

where $t$ is the fractional year and the coefficients are $a_0 = 72134.3$, $a_1 = -73.6115$, $a_2 = 0.01886$, $a_3 = 5.47721$, $a_4 =$
0.64527 and $a_5 = 1.12893$ ($r^2=0.95$, $p<0.001$). Due to the lack of local observations, this estimate does not include an
urban effect (local increase in atmospheric $pCO_2$) around the Halifax Harbour.

### 3.3 Validation of model physics

A multiyear simulation was initialized from BBMP data for Halifax Harbour and GLORYS and climatological data
for the shelf (Jan. 1, 2015) and spun up for 1 year. The period 2016-2023 was used for comparison with surface and
in-situ observations on the shelf for ROMS-H1 and at the compass buoy for ROMS-H2 and ROMS-H3 (Section 5.2.4).
Simulated surface temperature in ROMS-H1 agrees well with observed seasonal patterns over the shelf ($r^2>0.78$) and
at the 3 open boundaries of ROMS-H2 ($r^2=0.99$) (Figure S3, Table 1). The only discrepancy not captured by the model
is the large high-frequency variability in fall/winter surface temperature. Surface and in-situ temperature (Figure S4)
and salinity (Figure S5) are also in good agreement with the CTD profiles from Station 2 on the shelf ($r^2=0.77$ and
0.74, respectively, Table 1). At the Compass Buoy station in Bedford Basin, surface temperature (Figure S10) and
salinity (Figure S11) simulated with ROMS-H2 are close to observations, with some overestimation in spring and
summer for temperature and a slight underestimation for salinity. Vertical profiles agree well with observed
temperature, although the agreement is not as good for salinity ($r^2=0.78$ and 0.46, respectively, Table 1). At the surface,
temperature is somewhat overestimated in spring and summer, while salinity is slightly underestimated. The profile
comparisons indicate that the biases at the surface may be driven by the too shallow pycnocline. ROMS-H2 also tends
to overmix in the fall (Figure S10).

Temperature and salinity profiles at Compass Buoy site indicate that the model is able to reproduce the timing
of most of the discontinuities in water column properties, driven by convection in winter and intrusions of shelf waters
into the Bedford Basin in summer and fall (Garcia Larez, 2023). Sharp and concomitant increases in deep water
temperature and salinity at this site are a typical signal of intrusions events (Figure S10 and Figure S11). Despite the
agreement in timing, the magnitude of some events is overestimated by the model. The sharp surface pycnocline
(Figure S10), which likely enhances the estuarine circulation, may play a role in this mismatch.

Overall, the circulation model simulates the physical dynamics of the Halifax Harbour – Scotian Shelf system
well (Table 1). Thus, we deem the model as an appropriate framework for simulating the dynamics of oxygen and
carbonate system parameters in this area.

### 3.4 Shelf biogeochemistry



ROMS-H1 used the explicit biogeochemical model described in Section 3.1 but only oxygen and carbon chemistry were activated in the higher-resolution ROMS-H2 so we focus on these variables hereafter. Simulated oxygen compares well with the seasonal patterns observed at Station 2 ($r^2$=0.74, Table 1, Figure S6). The main difference is that the model overestimates deep oxygen concentrations (below ~100 m) and underestimates, at times, surface oxygen concentrations during the spring bloom. Despite these mismatches, oxygen variability is well represented in the model.

In Halifax Harbour, observations of alkalinity are strongly correlated with salinity (Atamanchuk et al., 2024; Figure S9). Since the initial and boundary conditions for alkalinity and salinity in ROMS-H1 have different sources (climatology and GLORYS, respectively), the simulated alkalinity (TA) versus salinity (S) relationship does not match the relationship observed in Halifax Harbour. Therefore, to calculate initial and boundary conditions for alkalinity in ROMS-H2 the following observed relationship from the Halifax Harbour was used:

$$TA_{fit} = 63.417 \times S + 235.135 \text{ mmol m}^{-3} \tag{2}$$

To avoid an artificial disequilibrium in pCO$_2$ (due to modified TA but not DIC), initial and boundary conditions for DIC were also recalculated with CO2SYS using simulated salinity, simulated pCO$_2$ and TA$_{fit}$. These calculated TA and DIC values are in good agreement with observations at Station 2 (Figure S7 and Figure S8, $r^2$=0.84 and 0.85, respectively, Table 1). The seasonal variability of surface DIC is also well captured (Figure S8). Similar to oxygen, deep TA (below ~100 m), and somewhat DIC, were generally overestimated.

**4    Reduced complexity biogeochemical model**

The model is based on the explicit biogeochemical model used in ROMS-H1 (Fennel, 2010; Fennel et al., 2006, 2013; Laurent et al., 2012, 2017, 2021) but simplified to improve computational efficiency for studying the carbonate system in the context of alkalinity enhancement. Since oxygen is measured routinely, in contrast to carbonate variables, and has sources and sinks similar to DIC (with opposite sign) the model includes oxygen as a diagnostic variable. Reduced

complexity biogeochemical models have previously proven useful in studies of hypoxia in Chesapeake Bay (Scully, 2013) and in the northern Gulf of Mexico (Yu et al., 2015).

**4.1    Model description**

The model state variables are TA, DIC and oxygen (O$_2$; Figure 2). The general model equations are available in the supplement information. The main differences to the original biogeochemical model used in ROMS-H1 are the

removal of state variables for nutrients (NO$_3$, PO$_4$, NH$_4$) and particulate organic matter (phytoplankton, zooplankton, small and large detritus) and their replacement with parameterizations of primary production and respiration in water column and sediment.

The ROMS-H2 domain includes the inner Scotian Shelf, where biological conditions are typical of those on the northwest North Atlantic shelves with biannual blooms, and the inner Halifax Harbour (including the Bedford

Basin) where conditions are highly eutrophic due to the input of anthropogenic nutrients from wastewater treatment



plants. Primary production and respiration are therefore parameterized differently in the harbour and on the shelf. The harbour parameterizations are also used in ROMS-H3.

### 4.1.1 Primary production

A multiyear simulation (2017-2022) was run for ROMS-H2 using the same explicit biogeochemical model (ROMS-
H2-BGC) as in ROMS-H1. The results were used to calculate time varying, but spatially averaged climatologies of surface primary production for the Scotian Shelf and Halifax Harbour (Figure S1). We then parameterized primary production on the Scotian Shelf ($PP_S$) and in Halifax Harbour ($PP_H$) with a fit to climatology such that it is a function of time:

$$PP_x(t) = a_1 + b_1 \times (\cos(b_2 t - 1) + 1))^{b_3} + (\sin(c_1 t - 1) + 1)^{c_1} + (\sin(d_1 t - 1) + 1)^{d_2} \qquad (3)$$
$$+ (\sin(e_1 t - 1) + 1)^{e_2}$$

where $x$ represents the Halifax Harbour ($PP_H$) or the shelf ($PP_S$), $t$ is the fractional year and PP has units of mmol $O_2$
$m^{-3}$ $d^{-1}$. These parameterizations are applied over the upper 30 m on the shelf and 15 m in Halifax Harbour to account for the higher light attenuation in the harbour (Craig et al., 2012). The coefficients for each location are listed in supporting Table S2. The parameterizations are compared to climatology in supporting Figure S1.

### 4.1.2 Water column respiration

The spatially and temporally averaged water column respiration in ROMS-H2-BGC for the shelf ($WR_S$;
mmol $m^{-3}$ $d^{-1}$) and the inner harbour ($WR_H$; mmol $m^{-3}$ $d^{-1}$) areas were approximated with a temperature dependency to parameterize water column respiration, such that:

$$WR_x = a_x \times \theta_T \qquad (4)$$

where $x$ represents the Halifax Harbour ($WR_H$; $a_H = 1.39403$) or the shelf ($WR_S$; $a_S = 1.31349$), and the temperature (T; ˚C) dependency $\theta_T = 0.59 \times 1.066^T$, as in the explicit biogeochemical model (Laurent et al., 2021). We assume
a respiratory quotient of 1 such that water column respiration has the same magnitude (with opposite sign) for $O_2$ and DIC.

### 4.1.3 Sediment respiration

On the shelf, sediment respiration ($SR_S$; mmol $m^{-2}$ $d^{-1}$) is parameterized as a temporally and spatially varying approximation of ROMS-H2-BGC results, such that:



$$SR_S = b_0 + c_0 lon + d_0 lat + \sum_{i=1}^{5} (e_0^i x_i + e_1^i x_i^2 + e_2^i x_i^3) \qquad (5)$$

where *lon* is longitude, *lat* is latitude, and $x_i$ represents bathymetry ($i$=1), surface and bottom temperature ($i$=2,3), and surface and bottom salinity ($i$=4,5). The coefficients for each variable are listed in supporting Table S2. As for water column respiration, we assume a respiratory quotient of 1. For the inner harbour, sediment flux measurements at 6 stations (Bedford Basin and Narrows) in fall 2022 and spring 2023 show a relationship between dissolved inorganic carbon flux ($F_H^{DIC}$; mmol C m$^{-2}$ d$^{-1}$) and depth ($z$; m) such that sediment respiration (SR$_H$; mmol m$^{-2}$ d$^{-1}$) is:

$$SR_H = F_H^{DIC} = 3.37 \times e^{0.0033 \cdot z} \qquad (6)$$

Measurements in the Bedford Basin showed a decrease in oxygen flux at depth ($F_H^{O_2}$; mmol O$_2$ m$^{-2}$ d$^{-1}$) under hypoxic conditions and therefore an oxygen (O$_2$; mmol m$^{-3}$) dependency was added to the sediment flux such that:

$$F_H^{O_2} = SR_H \times (1 - e^{-O_2/30}) \qquad (7)$$

### 4.2 Biogeochemical model validation

ROMS-H2 is able to simulate the seasonal variations of oxygen at the Compass Buoy station, including the seasonal subsurface depletion driven by respiration and occasional renewal of deep water oxygen during summer and fall intrusions ($r^2$=0.55, Table 1, Figure S12). Surface oxygen (upper 1 m) is generally underestimated, particularly during the annual spring bloom and in summer. As mentioned in Section 3.3, a few of the simulated intrusions were not visible in the observations (e.g. in fall 2019), generating some mismatches between simulated and observed subsurface oxygen. Another source of discrepancy was the overestimation of vertical mixing in fall that increased subsurface and bottom oxygen. This overestimation is reduced in ROMS-H3 (see Section 5.2.4).

At the Compass Buoy station, carbonate system observations are available at the surface (~1-3 m) and 60 m depth for comparison with the model (Table 1, Figure S13 and Figure S14). Surface alkalinity and DIC are in good agreement with observations. Surface variability, largely associated with river discharge, is particularly well simulated. At 60 m, the dynamics differ for simulated DIC and alkalinity. DIC accumulates in summer due to respiration, whereas alkalinity remains relatively constant. As for oxygen, the DIC accumulation is interrupted during deep water renewal by intrusions or deep mixing events (Figure S14). The rate and magnitude of the DIC increase in bottom waters is in good agreement with observations (e.g., 2017-2018), indicating that respiration is parameterized appropriately. The occasional mismatches with observations can have at least two causes: an overestimation of vertical mixing that counteracts the accumulation of DIC in late fall, which is improved in ROMS-H3 (see Section 5.2.4), and a mismatch in the occurrence or effect of intrusion events. The latter is more pronounced in the second part of the simulation.

The magnitude of simulated alkalinity agrees with observations at 60 m (Figure S13) with some mismatches in late fall (i.e. 2019-2022). The observed increase in alkalinity likely originates from benthic alkalinity fluxes associated with alkalinity production in the sediment (e.g., sulphate reduction), which are not yet parameterized in the model.





pCO$_2$ at the Compass Buoy station (calculated from TA and DIC with CO2SYS) has similar patterns as DIC
(Figure S15). At 60 m, the rate of change of pCO$_2$ is close to the observations, similar to the variability in deep DIC

(accumulation, mixing/intrusions). At the surface, pCO$_2$ is within the observed range but the variability is larger in
the model, probably driven by rapid changes in the location of the Sackville River plume. Simulated surface pCO$_2$
also tends to be elevated in the fall compared to observations, which is likely associated with overmixing at this time
in ROMS-H2.

ROMS-H3 better represents vertical mixing in the fall (more stratification, Figure S20–Figure S21). With this

higher-resolution model deep oxygen and DIC do not level off as in ROMS-H2 and therefore are better represented
in late summer and fall (i.e. Figure S12, Figure S22 and Figure S14, Figure S24). Alkalinity is still underestimated at
depth (bias: -36.7 mmol m$^{-3}$), likely due to the missing source from the sediment in the model (see above).

The model-observation comparison presented above indicates that both ROMS-H2 and ROMS-H3 represent well
the background carbonate system dynamics in the Halifax Harbour.

**5    Alkalinity addition module**

To simulate the effect of alkalinity addition on the carbonate system, several tracers were added to the biogeochemical
model. The addition module was designed so that a single simulation can be used to calculate the difference between
the realistic addition case and counterfactual.

**5.1    Description**

The carbonate system model simulates TA and DIC as state variables and computes the CO$_2$ air-sea flux online at
every timestep. New tracers were added for $\Delta$TA (mmol m$^{-3}$) and $\Delta$DIC (mmol C m$^{-3}$), which represent the added
alkalinity and the resulting increase in DIC, respectively. The model includes a point source for feedstock addition
that can be set constant or to vary with time using a forcing file. Since the feedstock is often partly in particulate form
(delivered as a slurry), a tracer TA$_P$ is added to represent the particulate phase of the added feedstock (TA$_{in}$; mmol m$^{-3}$

$^3$), which dissolves into $\Delta$TA and sinks at specified rates. The allocation of TA$_{in}$ into the dissolved ($\Delta$TA) and
particulate (TA$_P$) pools is set by the parameter $\theta_{P:D}$. This representation of dissolution of the particulate stock builds
on the formulation in Wang et al. (2025). All tracers are also subject to advection and mixing. To calculate air-sea gas
exchange with ($F_{CO_2}^a$; mmol C m$^{-2}$ d$^{-1}$) and without ($F_{CO_2}^c$; mmol C m$^{-2}$ d$^{-1}$) added alkalinity, surface pCO$_2$ is calculated
from TA and DIC (counterfactual case, no added alkalinity, pCO$_2^c$) and from TA+$\Delta$TA and DIC+$\Delta$DIC (case with

added alkalinity, pCO$_2^a$).

The formulations of the biogeochemical source and sink terms are as follows:

$$\frac{\partial TA_P}{\partial t} = \theta_{P:D} TA_{in} - k_{diss} TA_P - w_P TA_P \tag{8}$$

$$\frac{\partial \Delta TA}{\partial t} = (1 - \theta_{P:D}) \cdot TA_{in} + k_{diss} TA_P \tag{9}$$





where $k_{diss}$ (d$^{-1}$) is the dissolution rate and $w_P$ the sinking rate of TA$_P$. TA$_{in}$ > 0 at the addition location only.

At the surface, $\Delta$DIC varies due to the additional air-sea flux of $CO_2$ ($\Delta F_{CO_2}$; mmol C m$^{-2}$ d$^{-1}$) into the top layer as follows:

$$\left.\frac{\partial \Delta \text{DIC}}{\partial t}\right|_{z=0} = F_{CO_2}^a - F_{CO_2}^c \tag{10}$$

$$F_{CO_2}^c = vk_{CO_2} K_{CO_2} \left(\text{pCO}_{2,\text{air}} - \text{pCO}_2^c\right) \tag{11}$$

$$F_{CO_2}^a = vk_{CO_2} K_{CO_2} \left(\text{pCO}_{2,\text{air}} - \text{pCO}_2^a\right) \tag{12}$$

where $K_{CO_2}$ (mmol C m$^{-2}$ atm$^{-1}$) is the solubility of $CO_2$ and $vk_{CO_2}$ is the gas transfer velocity of $CO_2$. The gas transfer velocity (cm hr$^{-1}$) is parameterized using the gas transfer relationship in Wanninkhof (2014):

$$vk_{CO_2} = 0.251 \cdot u_{10}^2 \sqrt{\frac{660}{Sc_{CO_2}}} \tag{13}$$

where $Sc_{CO_2}$ is the Schmidt number of $CO_2$ and $u_{10}$ (m s$^{-1}$) is the wind velocity at 10 m height. The Schmidt number varies with temperature according to Wanninkhof (2014). The sensitivity of uptake to air-sea gas exchange was tested in (Laurent et al., 2024b).

In the bottom layer $N$, a loss term is added to mimics the incorporation and loss of feedstock into the sediment so that:

$$\left.\frac{\partial \Delta \text{TA}}{\partial t}\right|_{z=N} = (1 - \theta_{loss}) \cdot k_{diss} \text{TA}_P \tag{14}$$

where $\theta_{loss}$ is the fraction of TA$_P$ lost to the sediment.

At the release location an additional source term represents the feedstock addition so that $\theta_{P:D} \cdot \text{TA}_{in}$ is added to the TA$_P$ pool and $(1 - \theta_{P:D}) \cdot \text{TA}_{in}$ to $\Delta$TA.

### 5.2 Alkalinity addition simulations

ROMS-H2 was run for 1.5 years (July 2017 to December 2018) with constant addition in the first two months (60 days) at a rate of 1.29 mol s-1 for a total alkalinity addition of 6665·10$^3$ mol (Figure S2). This corresponds to 194 t of magnesium hydroxide (Mg(OH)$_2$) and approximately 250 t of brucite (including impurities), the mineral added in Halifax Harbour in 2023 (Cornwall, 2023). The model ran in six different configurations: three outfall locations in the inner (Mill Cove, location of a waste water treatment plant), mid (Tufts Cove, location of a cooling pipe from power plant) and outer (Herring Cove, location of a waste water treatment plant) harbour (Figure 1d) each with two different feedstocks, either fully dissolved or fully particulate. We assumed a particulate feedstock with a particle size of 12 μm, dissolution rate of 0.015 d$^{-1}$ and sinking rate of 5.5 m$^{-1}$. The dissolution and sinking characteristics of the



particulate feedstock were chosen to be similar to the brucite used in Halifax Harbour in 2023. The addition occurred via a point source and distributed uniformly over the upper 5 m. The cases with addition inside the harbour (Tufts Cove and Mill Cove) were repeated with the higher-resolution ROMS-H3 (Section 5.2.4). For analysis, we used 5 areas for spatial integration (Figure 1d), namely Bedford Basin (BB; 17.3 km$^2$), Inner Harbour (IH; 8.2 km$^2$), Outer Harbour (OH; 8.1 km$^2$), Northwest Arm (NA; 2.6 km$^2$) and Shelf (1785.9 km$^2$).

### 5.2.1 Tracking the dispersion and effects of alkalinity addition

We first describe the simulation with addition of dissolved feedstock at Tufts Cove (mid harbour). The sensitivity to feedstock and release location will be discussed in the following sections.

The location, magnitude and temporal evolution of ΔTA (Figure 3–Figure 5) indicate the fate of added alkalinity. Tufts Cove is located on the northern side of The Narrows where circulation is controlled by tides, wind, and freshwater discharge from the Sackville River (Shan et al., 2011). Depending on the conditions (tide phase, wind direction, river discharge) ΔTA is advected either into Bedford Basin toward the head of the estuary or the outer harbour and the open shelf (Figure 5). The average surface concentration of ΔTA in the harbour increased to ~10 mmol m$^{-3}$ by the end of the release (late August 2017) and dropped to <5 mmol m$^{-3}$ a month later (Figure 5). Once the release had stopped, ΔTA was mainly located in the Bedford Basin where residence time is longer (Wang et al., 2025). The Bedford Basin is also where the largest fraction of ΔTA was found during the simulation (Figure 4). After the release, it took about 4 months to flush most of ΔTA out of the harbour (Figure 4). Once advected to the inner shelf, ΔTA left the model domain rapidly. ΔDIC gradually increased at the surface during the release because of net ΔCO$_2$ uptake from the atmosphere (Figure 6). After the release, ΔDIC decreased rapidly and, as for ΔTA, remained longer in the Bedford Basin before being flushed out.

The spatial patterns of ΔF$_{CO_2}$ illustrate the location of the uptake. For the Tufts Cove addition, the magnitude of the uptake was largest in the Narrows, where the feedstock was released, followed by Bedford Basin, where ΔTA had a longer residence time (Figure 7). CO$_2$ uptake happened quickly for the dissolved feedstock with 44% of the total uptake within the model domain (59.2 tons) occurring during the 2 months of release and 95% (127.5 tons) within the first 6 months of the simulation (Figure 7 and Figure 8, orange line).

Maximum achievable net CO$_2$ uptake from the alkalinity addition was estimated to be 261 tons using a theoretical uptake efficiency of 1.34 g CO$_2$ per g Mg(OH)$_2$. The calculation assumed a potential CO$_2$ uptake of 0.89 mol CO$_2$ per mol TA, derived using averaged local properties in ROMS-H2. Total uptake over the course of the simulation was 134.4 tons so the realized uptake within the ROMS-H2 model domain was 51.5% of the achievable uptake (Tufts Cove/dissolved case, Figure 9). As discussed above, uptake varied spatially. Since the Inner Harbour (including The Narrows) is relatively small compared to the adjacent areas, total uptake in this area amounted to only 21.3%. The largest fractions of the total uptake were in the Outer Harbour (30.7%) and Bedford Basin (29.1%). Despite its large area, the shelf accounted for only 17.6% of the total uptake. The Northwest Arm had a negligible contribution (1.4%).

### 5.2.2 Effect of release locations





Next, we describe the effect of release location for the simulations with addition of dissolved feedstock. The sensitivity to feedstock type is discussed in the next section. The outcome of the alkalinity addition is sensitive to the release
location in the harbour and has potential implications when considering future deployments. For addition at the inshore site (Mill Cove), ΔTA concentration is much larger in the Bedford Basin even a month after the end of the release (Figure 4, see also Figure S16). The subsequent flushing is similar to the Tufts Cove case and most of the ΔTA was gone 4 months after the end of the release. For addition at the coastal site (Herring Cove), the amount of ΔTA is much lower (<5 mmol m$^{-3}$), most of it found near the release location in the Outer Harbour and on the shelf (Figure 4, see
also Figure S17). Since residence times in the outer harbour and on the shelf are much lower, ΔTA was transported rapidly out of the ROMS-H2 domain. Nearly all ΔTA was gone 1 month after the end of the release (Figure 4, see also Figure S17). Figure S18 (left panels) indicates the location of ΔTA transport out of the eastern and western boundaries.

The magnitude of the cumulative net $CO_2$ uptake is similar after 2 months of addition in the three simulations (Tufts Cove: 59.2 tons, Mill Cove: 64.8 tons, Herring Cove: 58.2 tons). The start of $CO_2$ uptake was somewhat delayed
in July 2017 (by ~2 weeks) in the Mill Cove release case but then followed the same trajectory as the other cases. Local conditions related to the influence of the Sackville River (shallow stratification, discharge pulses) may have limited the onset of $CO_2$ uptake at the beginning of the simulation. Subsequently, the difference between the three release cases was in the location of the uptake (Figure 7). The net $CO_2$ uptake was largest either in the Bedford Basin (Mill Cove release), the Narrows (Tufts Cove release) or the outer Harbour (Herring Cove release). After the initial
release period (September 2017 onward), the trajectories of net $CO_2$ uptake changed drastically between the three cases, which highlight the importance of residence time for the realized net $CO_2$ uptake. When alkalinity was added at the coastal site (Herring Cove), most of the uptake (73.0%) happened during the two months of feedstock release (92.6% was completed after the initial 3 months). ΔTA was dispersed rapidly out of the domain when the release stopped and therefore net uptake was limited then. With an inshore release (Mill Cove), 56.0% of the uptake occurred
after the initial two months (the release period).

Accordingly, spatially integrated net uptake (Figure 7) and the realized net uptake (Figure 9) diverged significantly for the three release sites. The realized net uptake was maximum for the inshore release (Mill Cove), reaching 65.1% by the end of the simulation, whereas only 30.5% of the achievable uptake was realized for coastal release site (Herring Cove). In all cases >95% of the uptake occurred within the first 6 months of the simulation. These
numbers are in line with the spatial redistribution of the uptake. For a release in Mill Cove, 70.2% of the uptake occurred inshore (Bedford Basin: 53.5%, Inner Harbour: 16.7%), whereas it was only 8.2% for a release in Herring Cove (Bedford Basin: 4.3%, Inner Harbour: 3.9%). In this latter case, 51.3% of the uptake was in the outer harbour (where alkalinity is released) and 38.8% on the shelf (Figure 9).

### 5.2.3 Effect of feedstock type

The addition of alkalinity as particulate material at the 3 locations resulted in a delayed availability of ΔTA, controlled by the feedstock's dissolution kinetics, in comparison to the dissolved feedstock addition (Supporting Figure S19, Figure 4). The spatial distribution of ΔTA was also modified because a large fraction of the particulate feedstock sank



to the bottom layer. There, it was entrained in the return flow of the estuarine circulation toward the Bedford Basin where it accumulated and dissolved (Figure 10). ΔTA was relocated toward the Bedford Basin in this case (Figure 3).

With particulate feedstock release at the 3 locations, only a small fraction of the uptake occurred during the release period (7.3–10.9%, Figure 7 and Figure 8). After 6 months, only 71.5-78.3% of the achievable uptake was realized and, for the releases at Mill Cove or Tufts Cove, it took the whole simulation (1.5 years) to reach uptake values close to those in the case with fully dissolved feedstock (93.1–97.7%, Figure 7 and Figure 9). These delays are the result of feedstock dissolution as well as the time it takes for the ΔTA signal to re-emerge at the surface and

produce $CO_2$ uptake. The spatial distribution of net $CO_2$ uptake did not change from the dissolved feedstock case for the Mill Cove release but a relocation towards the Bedford Basin was observed for the Tufts Cove release case (Figure 9).

For a release at Herring Cove, total uptake by the end of the simulation was much lower for particulate than dissolved feedstock (53.1 tons, 66.6% of dissolved case). When particulate feedstock is released at Herring Cove,

material that is not entrained in a return flow towards the head of the harbour is moved downcoast by the coastal current on the shelf and ultimately leaves the domain relatively quickly (Figure S18). In this case, the potential uptake is not realized within the model domain.

### 5.2.4  High-resolution Halifax Harbour model

The capability of the addition model was tested above within ROMS-H2 which has 150-m horizontal resolution in the harbour and on the shelf. The third level nest ROMS-H3, with 51 m resolution, is better able to represent small scale features in the harbour. The high resolution also improves simulation of vertical mixing (Wang et al., 2025). ROMS-H3 was run for the same period as the addition experiments in ROMS-H2 (July 2017 to December 2018).

Alkalinity addition in ROMS-H3 was tested for the four cases where release occurred inside ROMS-H3,

namely Tufts Cove and Mill Cove each with either dissolved or particulate feedstock. Boundary conditions at the entrance of the harbour were provided by the corresponding ROMS-H2 simulations (see Sections 5.2.2 and 5.2.3). ROMS-H3 simulates small scale features in the harbour, driven by local wind and tides, especially around the release location (Figure 11). High resolution was particularly important to represent spatial features in The Narrows for the Tufts Cove release (Figure 11b) and in Bedford Basin for the Mill Cove release (Figure 11c). The dispersion of

feedstock outside the harbour was tracked, as previously discussed, with ROMS-H2 (Figure 11a,d).

Net $CO_2$ fluxes in the three-level nest were consistent with ROMS-H2 (Figure S25), especially for a release from Tufts Cove (Figure S26a,b). The main temporal difference occurred with particulate feedstock during a period of intrusion/mixing from late December 2017 to January 2018. In Bedford Basin, the intrusion brought to the surface deep ΔTA (dissolved from particulate feedstock in the bottom layer) and was followed by a period of enhanced net

$CO_2$ fluxes in ROMS-H3. The differences were more pronounced with a release at Mill Cove (Figure S26c,d). For dissolved feedstock, net $CO_2$ flux was larger in late summer and fall, consistent with the better representation of vertical mixing in ROMS-H3 (Figure S26c). For particulate feedstock, the net $CO_2$ flux was also higher following the intrusion (Figure S26d). Integrated in time, total net $CO_2$ flux did not change significantly with ROMS-H3 for a release



at Tufts Cove. For Mill Cove, net $CO_2$ flux increased only slightly with particulate feedstock, whereas the realized
uptake increased to 68.8% with dissolved feedstock (+3.1%). This increase is associated with enhanced fluxes in the
Bedford Basin (Figure S25).

## 6 Discussion

### 6.1 Model performance

The circulation model is modified from the nested model of Wang et al. (2025) using rotated and refined grids. Not
surprisingly, the physical model had similar performance when compared with observations (Table 1). The main
difference to Wang et al. (2025) and earlier circulation models of the Halifax Harbour (Shan, 2010; Sui et al., 2024)
was the coupling with biogeochemistry. The simplified biology, based on observed respiration rates in the Bedford
Basin and simulated rates on the shelf, was sufficient to reproduce the characteristic benthic respiration-driven deep-
water depletion/accumulation of oxygen and DIC in the Bedford Basin (Burt et al., 2013; Rakshit et al., 2023, 2025).
The model also simulated the gradient in oxygen and inorganic carbon variables between the harbour and the shelf as
seen when comparing the time series at the Compass Buoy station and Station 2. Using parameterizations rather than
full descriptions of biological sources and sinks allowed for efficient carbonate system dynamics in Halifax. In this
context the simplified model, which is meant to represent the background state and counterfactual case during TA
addition experiments, seems fit for purpose.

Winter mixing and water intrusions from the shelf (Garcia Larez, 2023; Sui et al., 2024) are the two physical
processes controlling subsurface water renewal in Bedford Basin (Burt et al., 2013; Haas et al., 2021; Rakshit et al.,
2023). The two processes are important in the context of alkalinity enhancement because they lower residence time
(Wang et al., 2025) and bring subsurface waters into contact with the atmosphere. Deep water renewal from convective
mixing and intrusions are simulated well by the model, with some discrepancies to observations. More specifically,
the frequency and magnitude of intrusions are sometimes overestimated in the baseline simulation (see Garcia Larez,
2023 for a detailed statistical comparison with the Wang et al. model version). OAE simulations were carried out over
a period when convective mixing and intrusions were well represented in both ROMS-H2 and ROMS-H3 so their
background dynamics can be assumed correct. The impact of intrusion events is illustrated, for example, by the
enhanced net $CO_2$ uptake from December 2017 to January 2018 in the case with particulate feedstock where an
470 intrusion redistributed ΔTA from the bottom layer in late December (Figure 8).

Validating the simulated circulation and background carbonate chemistry is an important step prior to
simulating alkalinity additions in the harbour as it gives confidence in the model results. Local circulation controls the
distribution of added alkalinity, whereas background carbonates can influence the timing of net $CO_2$ uptake (Fennel,
2025). However, other coastal systems where the model may be relocated won't necessarily have long-term time series
475 that can be used for validation. In such case, available information about general circulation features, satellite SST
and any prior data should be used. Comparison with climatological carbonate data or targeted sampling are needed to
increase confidence in the model outputs.

### 6.2 Alkalinity addition module



The model with alkalinity addition module was developed with the goal of supporting realistic OAE field trials and associated research and operations in Halifax Harbour (Laurent et al., 2024a) as well as to provide more general insights into the potential of coastal OAE, the optimal design of field operations, and as a flexible and freely available capability for MRV in this and other coastal systems. The key characteristics of the addition module are: 1) its capability to simulate realistic time series of point source addition, 2) tuneable feedstock properties (i.e. dissolved and particulate fractions, dissolution and sinking rates), 3) spatially and temporally resolved tracking of feedstock effects on the carbonate system, 4) quantification of the additional $CO_2$ air-sea flux using a unified code for addition and counterfactual cases, and 5) relocatability with, arguably, minor effort.

The capability for point source addition was included in the model to enable time-varying rates of feedstock dosing rates, which is relevant for real-world operations. This level of detail is especially important in dynamic tidal environments, such as the Halifax Harbour, where the timing of the release with respect to the tidal phase influences the spatial dispersion pattern of the feedstock (Wang et al., 2025). Fine-scale dispersion features, as seen in ROMS-H3 (Figure 11) depend on the accurate timing of additions. They also depend on the characteristics of the feedstock. Many types of alkaline material are available for OAE (Caserini et al., 2022; Eisaman et al., 2023; Renforth, 2019) with a range of specific properties (e.g., dissolution kinetics, particle size). Wang et al. (2025) showed that feedstock properties have a strong effect on exposure and detectability of OAE signals in Halifax Harbour. Building on this prior work, the addition module presented here shows examples of the effects of feedstock properties on the carbonate system and net $CO_2$ uptake in the Halifax Harbour (Figure 8 and Figure 9; see Section 6.3 below).

Coupled, high-resolution, spatially resolved regional models are essential for MRV of OAE (Fennel et al., 2023). They provide the counterfactual case that simply cannot be obtained with in-situ measurements (Fennel, 2025). The simulations presented here show that for a release in Halifax Harbour, depending on the feedstock and release location, 20–69% of the potential $CO_2$ uptake was realized inside or in direct surroundings of the harbour (Figure 9), mostly within six months of release (Figure 7 and Figure 8). One can expect comparable results for realized uptake in other coastal systems with similar characteristics. Residence time is a particularly relevant factor (Wang et al., 2025). Despite their importance, inshore/coastal areas are poorly resolved, if at all, in global ocean models (Laurent et al., 2021) and their residence time can be underestimated significantly in coarse models (Rutherford and Fennel, 2018). Global models therefore may be more appropriate for assessing first-order, large-scale patterns of coastal OAE potential and for simulating far-field and global responses of the Earth system to coastal additions (Feng et al., 2017; He and Tyka, 2023). Regional models, such as the one presented here, should be deemed essential for a comprehensive verification of net $CO_2$ uptake resulting from coastal additions. Our results indicated that the multi-nest ROMS model with TA addition module has the appropriate level of skill and thus capability for that task. The two or three nested levels provided sufficient resolution for representing the background state of the carbonate system and for assessing additionality at local to regional scale.

One of the features of the addition module is its unified source code for the paired, simultaneous simulation of the addition and counterfactual cases providing computational efficiency and simplicity. Zhou et al. (2025) took a similar approach in their global model. Adding ΔTA, ΔDIC and particulate feedstock as state variables in the addition module provides a simple and efficient way to track OAE-related perturbations of the background carbonate system.



An optional, diagnostic state variable was added to keep track of the alkalinity originating from in-situ dissolution of feedstock in contrast to feedstock that was added directly in dissolved form. It should be relatively easy to relocate the framework presented here to other coastal systems. At a minimum, relocation requires knowledge of the local alkalinity to salinity relationship and local sources and sinks of alkalinity and DIC from rivers, sewage, and sediments.
Ideally, information would also be available to tune the parameterizations of primary production and water-column and sediment respiration. Feedstock should also be characterized well.

### 6.3 Simulating additionality in the Halifax Harbour

The detectability of $\Delta$TA during addition operations and the quantification of its effects on air-sea fluxes is a challenge for OAE (Ho et al., 2023). This is particularly true in systems with large seasonal variations of the background
carbonate system variables such as Halifax Harbour (Figure S13–Figure S15). In this system, Wang et al. (2025) demonstrated that detectability varies in time and space and provided insights for maximizing $\Delta$TA detection while minimizing exposure risk. Our results on the distribution of $\Delta$TA are in line with these findings. The mean estuarine circulation of Halifax Harbour results in retention of the $\Delta$TA signal in the Bedford Basin. This area may be an ideal spot for maximizing the detectability of $\Delta$TA and its effects on $CO_2$ uptake although, as noted by Wang et al. (2025),
increased detectability goes hand in hand with an increased risk of ecosystem exposure. The latter would also be most acute if the release occurred at the head of the basin (Figure 10 and Figure 11).

The timescale of $CO_2$ equilibration is an important parameter for OAE because it influences the location of net $CO_2$ uptake as well as the maximum achievable uptake if the residence time of $\Delta$TA in the surface mixed layer is shorter than the equilibration timescale. In the open ocean, the timescale of $CO_2$ equilibration is 3 to 6 months on
average (Jones et al., 2014). Following an alkalinity addition, the time scale will vary with the thickness of the surface layer and the initial disequilibrium with the atmosphere (Fennel, 2025). In the simulations presented here, the time evolution of net $CO_2$ uptake is consistent with these previously reported timescales. Note that residence time is relatively similar to the equilibration time scale in the entire Bedford Basin (*e*-folding flushing time ~90 days, relevant for the particulate feedstock), smaller in the upper Bedford Basin only (0-20 m, *e*-folding flushing time ~39 days,
relevant for the dissolved feedstock) but much shorter elsewhere (<5 days) (Shan and Sheng, 2012; Wang et al., 2025).

Release location is a main factor controlling $\Delta$TA retention in the Bedford Basin (Figure 4) and the magnitude of net $CO_2$ uptake occurring in the harbour (Figure 7 and Figure *9*). A Mill Cove release resulted in the most uptake within the model domain, independently of the feedstock (166–170 t $CO_2$ uptake). 53% of uptake occurred in the Bedford Basin in this case. Another reason to carry out sensitivity simulations is to understand the effect of local
circulation features on the dispersion of feedstock. Wang et al. (2025) found that feedstock characteristics affect the residence time of the addition signal. We also found a redistribution of the $\Delta$TA signal, controlled by the estuarine return flow, toward the Bedford Basin (Figure 3) and a delay in the uptake due to the feedstock dissolution kinetics (Figure 9). However, despite the spatial and temporal redistribution of net $CO_2$ uptake, feedstock characteristics had a limited effect on total uptake by the end of the simulation when release occurred in the Bedford Basin (Mill Cove)
and the inner harbour (Tufts Cove).



Tufts Cove is the location for alkalinity addition for on-going OAE operations in the Halifax Harbour. This is a practical choice and as discussed above total uptake is significant in this case. Our simulations also provide insights into potential alternative addition sites at WWTPs, which have been suggested for cost effective operations (Cai and Jiao, 2022). Additional factors such as the differences between the outfall types, their discharge depth and operational constraints need to be considered in each case of the future proposed sites. Aside from Mill Cove, two WWTPs are in the inner harbour (similar to Tufts Cove) and two in the outer harbour, including Herring Cove (Figure 1d). Since feedstock disperses rapidly at a coastal release site, MRV would be more challenging for an outer harbour WWTP addition. In this case, a larger regional or global model would be essential to track net uptake.

The difference in horizontal model resolution between ROMS-H2 and ROMS-H3 (i.e. 150 m versus 50 m) did not have a systematic effect on net fluxes and therefore on the outcome of the additions. Nonetheless, higher resolution resulted in fine scale features of ΔTA and ΔDIC (Figure 11), knowledge of which are informative to guide field sampling, validate simulated feedstock dissolution and dispersion, and therefore provide confidence in the estimates of net $CO_2$ air-sea fluxes. ROMS-H3 better resolved vertical mixing in the Bedford Basin, which ultimately influenced the residence time of ΔTA in the Bedford Basin, as in Wang et al., (2025). The resulting relocation of net $CO_2$ uptake toward the Bedford Basin was small but not negligible for a release at Mill Cove (Figure S25).

### 6.4 Current limitations and future development

Several processes are either poorly constrained or not represented in the addition module as presented here. For instance, a constant sediment loss term that mimics the incorporation and loss of feedstock into the sediment is included in the addition module, but given the lack of observation-based constraints it was set to zero in our simulations. Earlier sensitivity tests indicated that results scale with the magnitude of this loss term (Laurent et al., 2024a, c). This parameterization provides a first-order representation of sediment loss but should be refined further to include possible feedback between feedstock deposition and benthic alkalinity flux (e.g., Bach, 2024). A better understanding of feedstock-sediment interaction will be necessary to simulate this feedback.

BBMP observations indicate a source of TA in the deep layer of Bedford Basin (Figure S13). Burt et al. (2013) and Rakshit et al. (2025) suggested the occurrence of sediment denitrification but they did not observe or report a matching TA flux. The source of TA accumulation at depth should be investigated and then parameterized accordingly in the model. This is the subject of an on-going study. Other sources of alkalinity and DIC were also poorly constrained in the model due to a lack of observations. For example, additional carbonate data should be collected in the Sackville River and WWTPs effluents, to better constrain natural sources of DIC and alkalinity in the model.

### 7 Conclusions

We developed a reduced-complexity biogeochemical module within a high-resolution nested model of the Halifax Harbour. The model was able to simulate the variability of the carbonate system parameters observed in the harbour. To simulate OAE, we included an option for alkalinity addition that tracks particulate feedstock, added alkalinity, and the resulting net $CO_2$ uptake through an additional DIC pool. The model provides the capability for testing the outcome



of OAE scenarios, to quantify additionality and realized uptake within the model domain, as well as to operationally support OAE field trials. It is fully relocatable, as a module within the ROMS framework. We tested the addition module for the Halifax Harbour to assess its capability and support OAE activities in the harbour. Release locations had a strong effect on the outcome of the addition, whereas feedstock types influenced the distribution of the signal.

The simulations showed that up to 69% of the net $CO_2$ uptake was realized within the model domain, most of it inside the harbour. These results stress the importance of operational design as well as the use of high-resolution regional models when quantifying additionality. In the future, the nested model will be applied to on-going OAE operations in the Halifax Harbour, to further validate the addition module, understand the fate of the alkalinity added to the harbour and to quantify net $CO_2$ uptake.

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

**Funding**

AL, BW, DA, and KF were supported by the Carbon to Sea initiative grant Ocean Alk-Align. AL, BW, and KF were also supported by the NSERC Alliance Missions grant ALLRP 570525-2021 and NSERC Discovery grant RGPIN-2022-688-02975.



# Tables

Table 1. Model-observations comparison statistics for figures S4–14

| Grid | Location | Variable | Type | R² | bias | RMSE |
|---|---|---|---|---|---|---|
| H1 | Shelf (maps) | SST (seasonal) | DJF | 0.981 | -0.15 | 0.23 |
| | | | JJA | 0.781 | -0.09 | 0.23 |
| | | | MAM | 0.940 | -0.12 | 0.21 |
| | | | SON | 0.944 | -0.10 | 0.14 |
| | Shelf (series) | SST (H2 boundaries) | South | 0.993 | 0.01 | 0.53 |
| | | | West | 0.989 | -0.01 | 0.66 |
| | | | East | 0.991 | 0.06 | 0.62 |
| | Station 2 | Temperature | CTD Profiles | 0.773 | 0.75 | 1.89 |
| | | Salinity | | 0.741 | 0.28 | 0.76 |
| | | Oxygen | | 0.738 | 0.19 | 30.3 |
| | | DIC* | Water samples | 0.853 | 11.0 | 40.3 |
| | | TA* | | 0.842 | 37.9 | 56.2 |
| H2 | BBMP | Temperature | CTD Profiles | 0.777 | 0.45 | 1.66 |
| | | Salinity | | 0.457 | -0.23 | 0.64 |
| | | Oxygen | | 0.554 | -19.2 | 65.2 |
| | | DIC | Water samples | 0.585 | 24.3 | 109 |
| | | TA | | 0.520 | 15.4 | 63.3 |
| | | pCO₂ | | 0.408 | 38.6 | 562 |

Seasonal averages: DJF: Dec.–Feb., JJA: Jun.–Aug., MAM: Mar.–May, SON: Sep.–Nov.
Units: Temperature: ˚C, Oxygen, DIC, TA: mmol m⁻³, pCO₂: µatm
* Corrected values based on the TA versus salinity relationship (see Section 3.4)



**Figures**

Figure 1. Study area (A, B) and nested model grids (C, D). A. Bathymetry on the Northwest North Atlantic (NWA; https://www.gebco.net, color scale as in C) including the southern Labrador Sea (LS), the Gulf of Saint Lawrence (GSL) and the Gulf of Maine (GoM). ROMS-H1 is delimited by the black rectangle. B. Satellite view of the Halifax Harbour (http://www.earth.google.com [September 17, 2024]). C. Bathymetry and outer limits of the nested grids. D. Bathymetry and outer limits of ROMS-H3 plotted on top of ROMS-H2. The red star indicates the location of the
compass buoy, the red dots the release locations at Mill Cove (MC), Tufts Cove (TC) and Herring Cove (HC) and the blue dots other WWTPs. The thin grey lines indicate the limits of the 5 areas used for spatial integration, namely Bedford Basin (BB), Inner Harbour (IH), Outer Harbour (OH), Northwest Arm (NA) and Shelf. Satellite image in panel B was generated with Google Earth. Image credits: Landsat/Copernicus, © 2025 Maxar Technologies, © 2025 CNES/Airbus.

Figure 2. Schematic of the biogeochemical model. Dark grey boxes represent the main variables of the biogeochemical model (control). Pink boxes and arrows indicate the addition model variables and processes, respectively.

Figure 3. Time-averaged (Jul 2017-Dec 2018), vertically integrated ΔTA in the harbour and surrounding areas during each simulation (excluding offshore areas where ΔTA~0).

Figure 4. Area-integrated time series of ΔTA in the simulations with dissolved and particulate feedstock for the four regions of the model domain. The Northwest Arm region is not shown because area-integrated ΔTA is small there. The spatial limits of the regions are indicated on Figure 1d (BB: Bedford Basin, IH: Inner Harbour, OH: Outer Harbour, NA: Northwest Arm).

Figure 5. Snapshots of surface ΔTA in the simulation with the release of fully dissolved feedstock from Tufts Cove.
Satellite image was generated with MapBox © Mapbox

Figure 6. Snapshots of surface ΔDIC in the simulation with the release of fully dissolved feedstock from Tufts Cove. Satellite image was generated with MapBox © Mapbox

Figure 7. Maps of time-averaged (Jul 2017-Dec 2018) net $\Delta CO_2$ uptake during the simulations (excluding offshore areas where $\Delta CO_2$ uptake ~0)

Figure 8. Time series of spatially integrated net $CO_2$ uptake in the simulations with dissolved (top) and particulate (bottom) feedstock.

Figure 9. Realized net $CO_2$ uptake for each simulation. The colors in the legend indicate the total uptake in each area of the model grid. Pink and olive-green lines under the Tufts Cove – dissolved bar indicate the realized uptake in ROMS-H3 (inside harbour) and in the outer area of ROMS-H2 (outside harbour), respectively (see Figure 1d).
Maximum achievable net $CO_2$ uptake from the alkalinity addition was estimated to be 261 tons using a theoretical uptake efficiency of 1.34 g $CO_2$ per g $Mg(OH)_2$ (see Section 5.2.1). The realized uptake is the relative difference between maximum achievable net uptake and simulated net uptake.

Figure 10. Average distribution of particulate feedstock during the simulations with addition at the three locations.

Figure 11. Snapshots of surface ΔTA in the nested simulations with ROMS-H2 and ROMS-H3 for a release of fully
dissolved feedstock at Tufts Cove (a,b) and at Mill Cove (c,d). Panels b and c show a zoom in on the inner Harbour (b) and the Bedford Basin (c). Satellite images were generated with MapBox © Mapbox



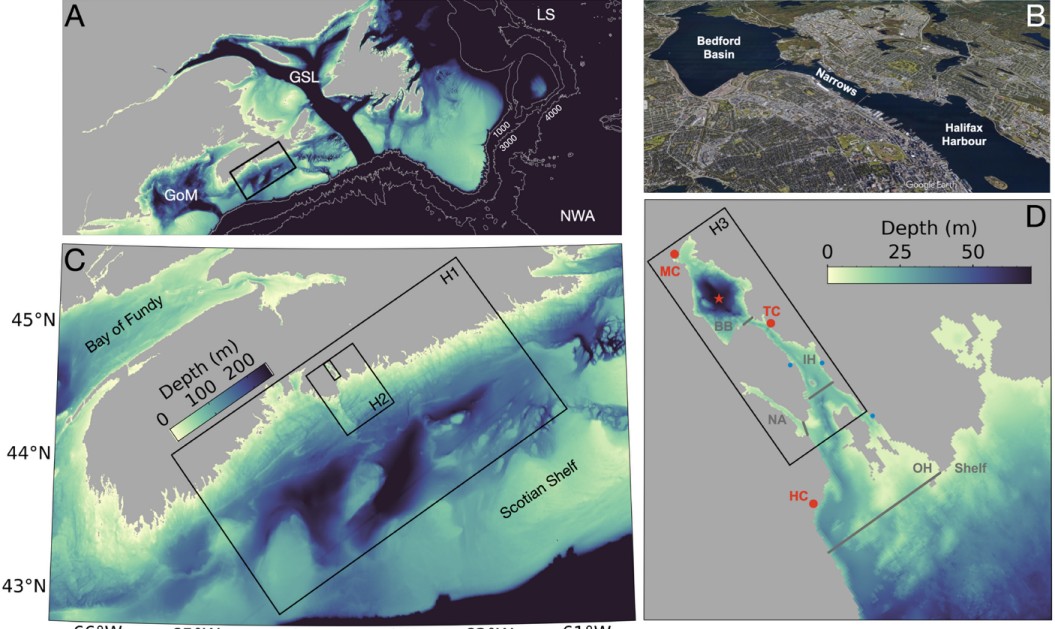

Figure 1. Study area (A, B) and nested model grids (C, D). A. Bathymetry on the Northwest North Atlantic (NWA; https://www.gebco.net, color scale as in C) including the southern Labrador Sea (LS), the Gulf of Saint Lawrence (GSL) and the Gulf of Maine (GoM). ROMS-H1 is delimited by the black rectangle. B. Satellite view of the Halifax Harbour (http://www.earth.google.com [September 17, 2024]). C. Bathymetry and outer limits of the nested grids. D. Bathymetry and outer limits of ROMS-H3 plotted on top of ROMS-H2. The red star indicates the location of the compass buoy, the red dots the release locations at Mill Cove (MC), Tufts Cove (TC) and Herring Cove (HC) and the blue dots other WWTPs. The thin grey lines indicate the limits of the 5 areas used for spatial integration, namely Bedford Basin (BB), Inner Harbour (IH), Outer Harbour (OH), Northwest Arm (NA) and Shelf. Satellite image in panel B was generated with Google Earth. Image credits: Landsat/Copernicus, © 2025 Maxar Technologies, © 2025 CNES/Airbus.



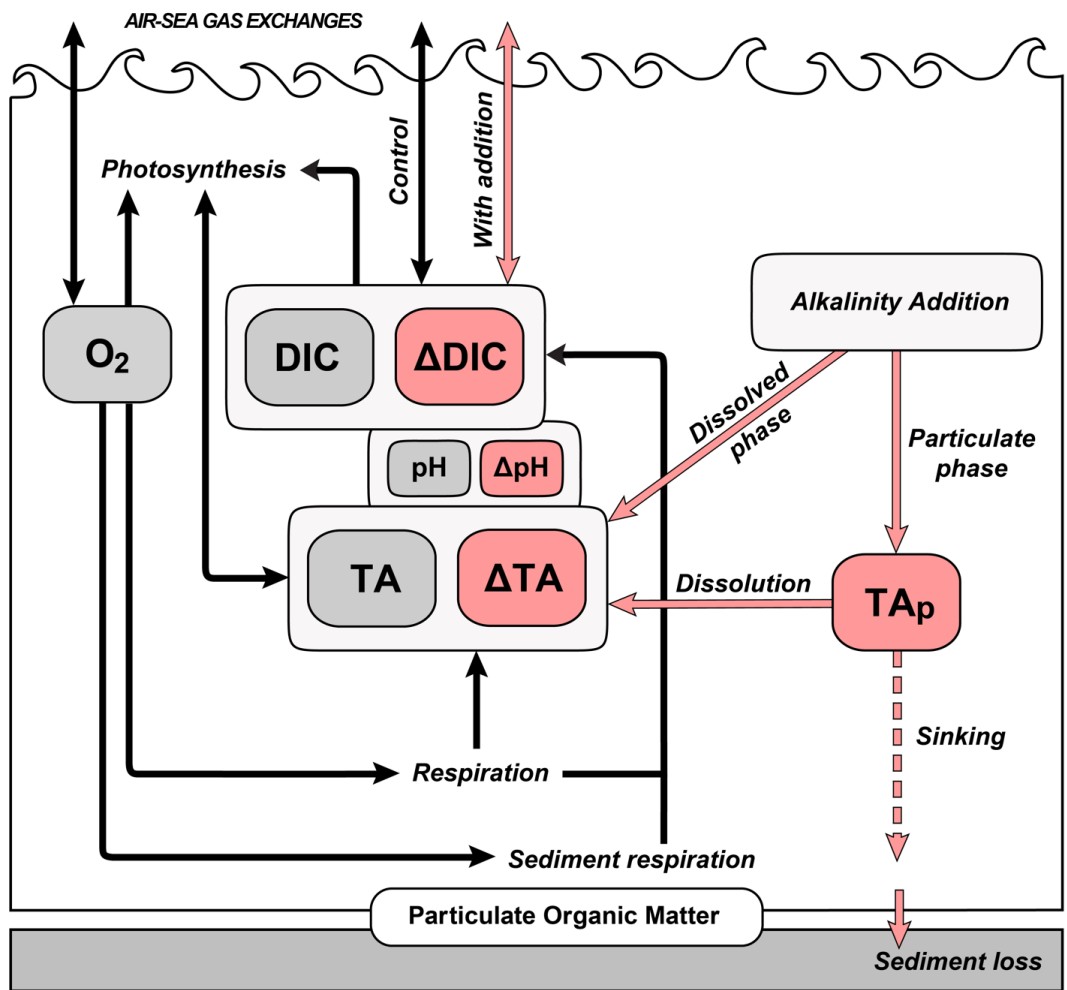

Figure 2. Schematic of the biogeochemical model. Dark grey boxes represent the main variables of the biogeochemical model (control). Pink boxes and arrows indicate the addition model variables and processes, respectively.






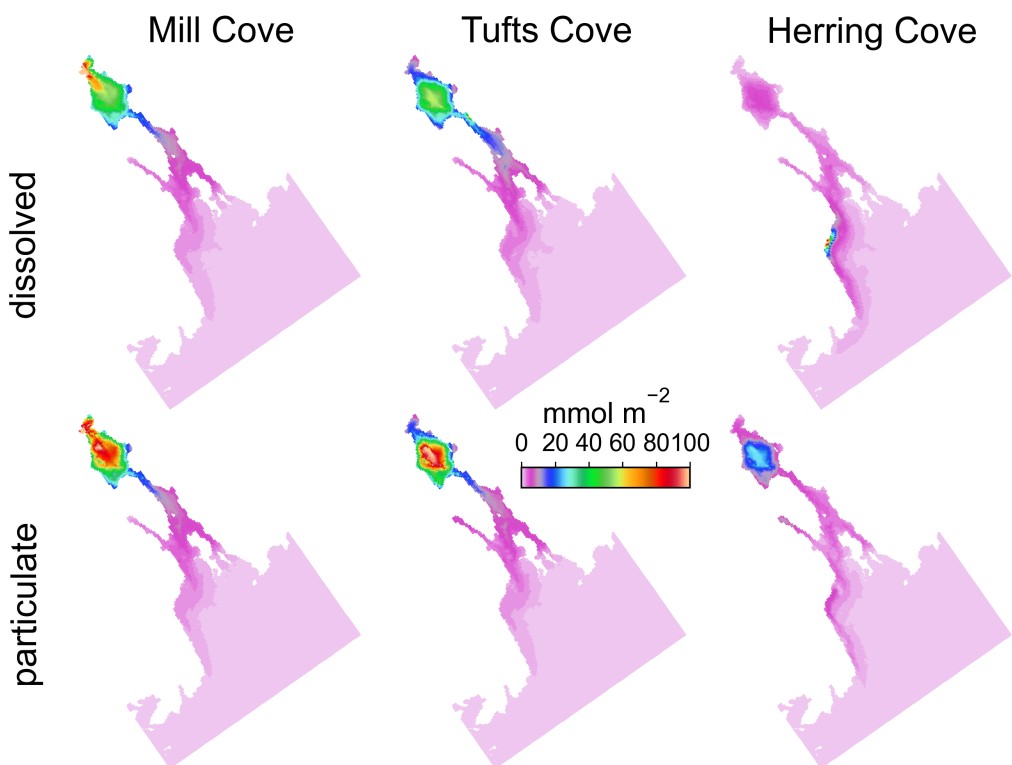

Figure 3. Time-averaged (Jul 2017-Dec 2018), vertically integrated ΔTA in the harbour and surrounding areas during each simulation (excluding offshore areas where ΔTA~0).




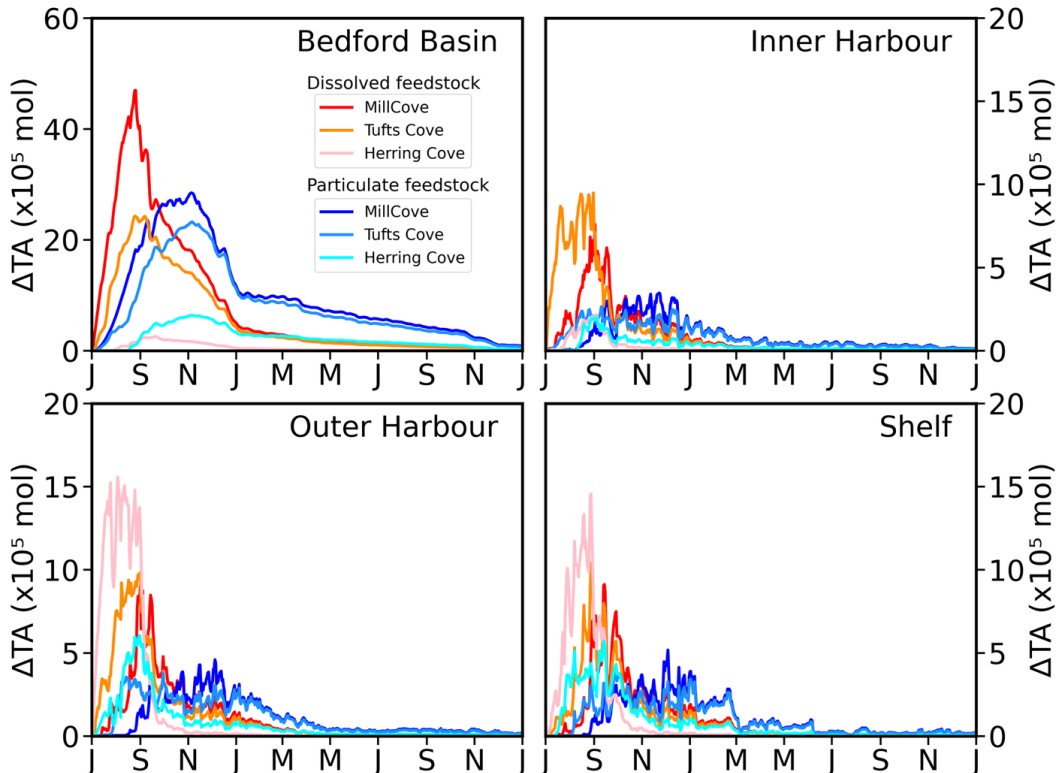

Figure 4. Area-integrated time series of ΔTA in the simulations with dissolved and particulate feedstock for the four regions of the model domain. The Northwest Arm region is not shown because area-integrated ΔTA is small there. The spatial limits of the regions are indicated on Figure 1d (BB: Bedford Basin, IH: Inner Harbour, OH: Outer Harbour, NA: Northwest Arm).


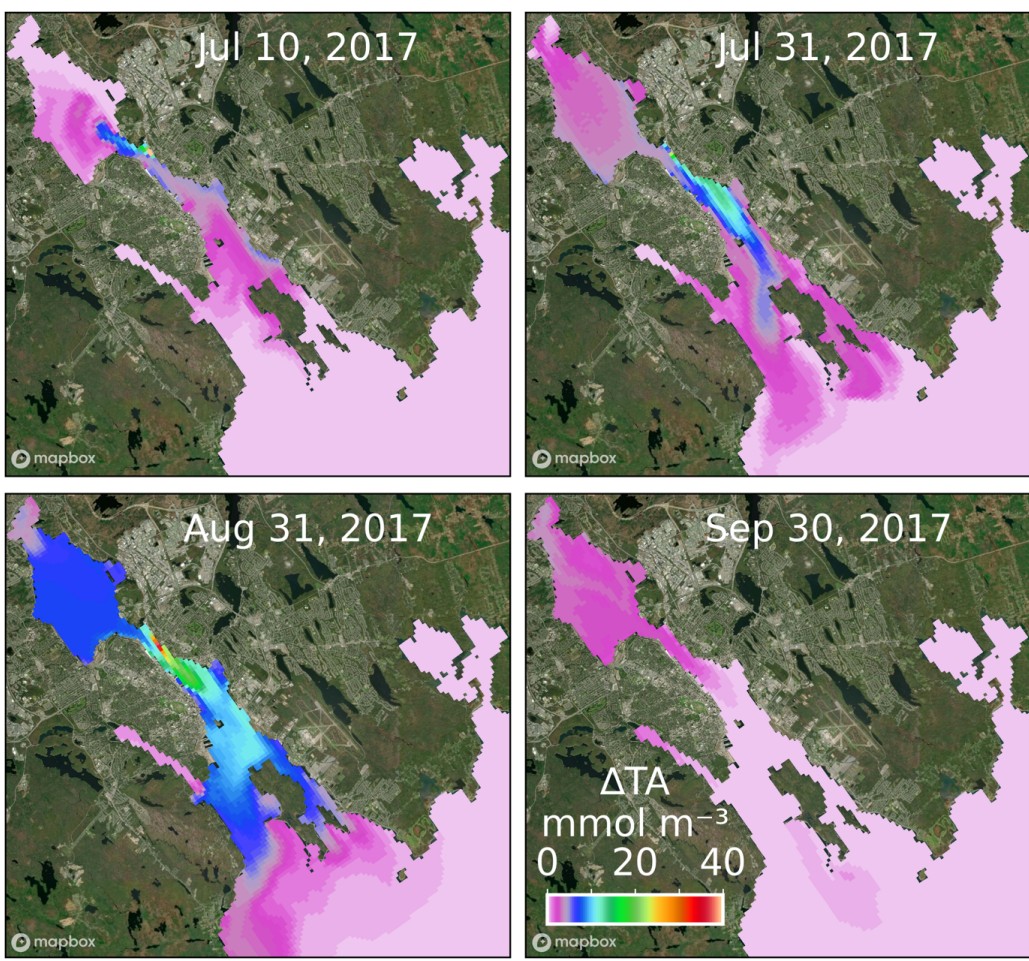

Figure 5. Snapshots of surface ΔTA in the simulation with the release of fully dissolved feedstock from Tufts Cove. Satellite image was generated with MapBox © Mapbox




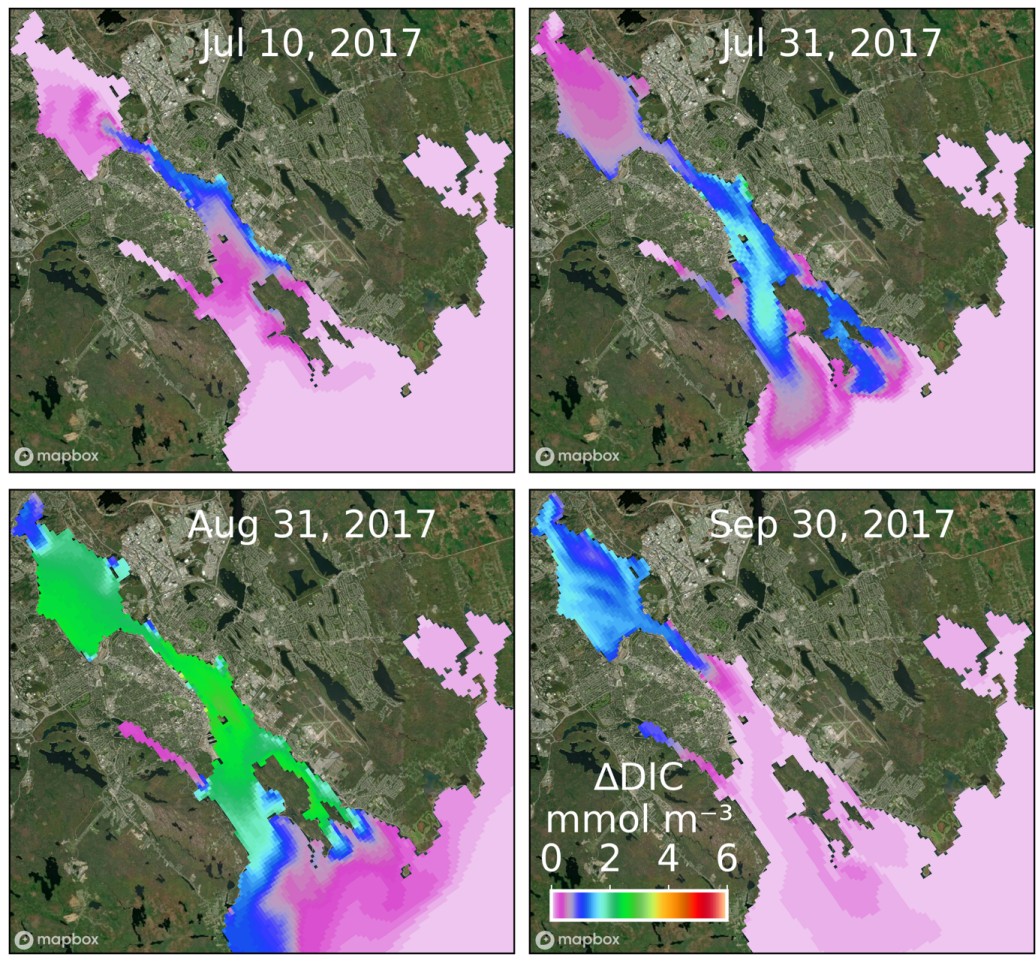

Figure 6. Snapshots of surface ΔDIC in the simulation with the release of fully dissolved feedstock from Tufts Cove. Satellite image was generated with MapBox © Mapbox




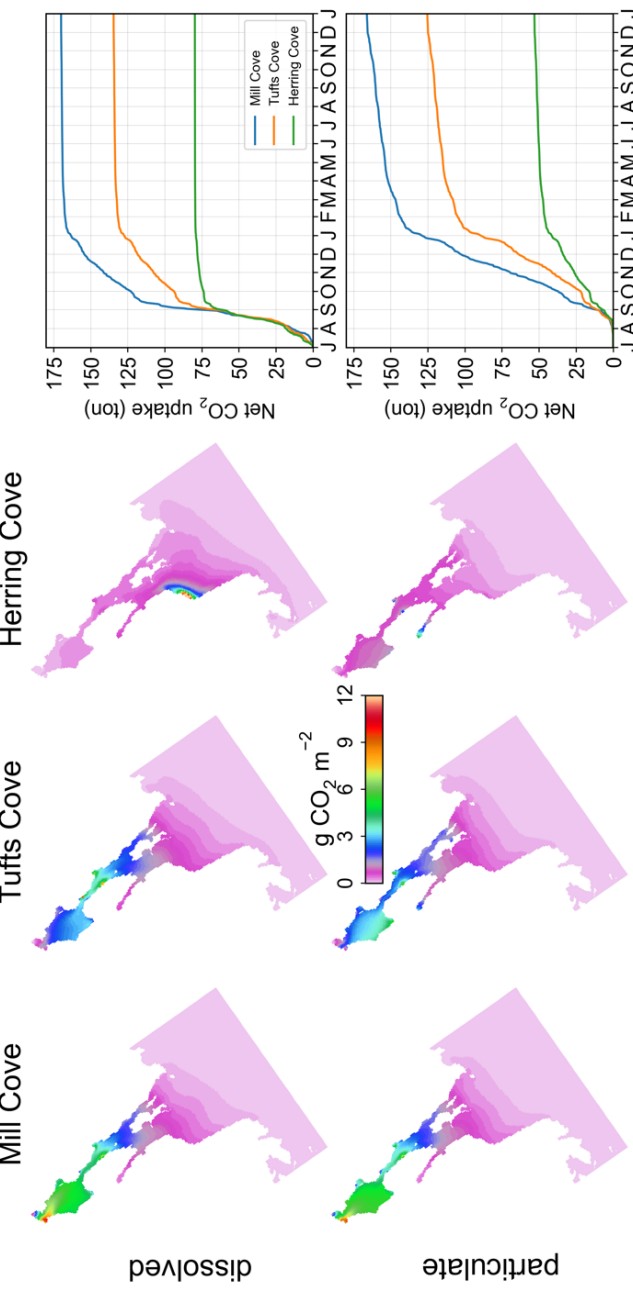

Figure 7. Maps of time-averaged (Jul 2017-Dec 2018) net $\Delta CO_2$ uptake during the simulations (excluding offshore areas where $\Delta CO_2$ uptake ~0)



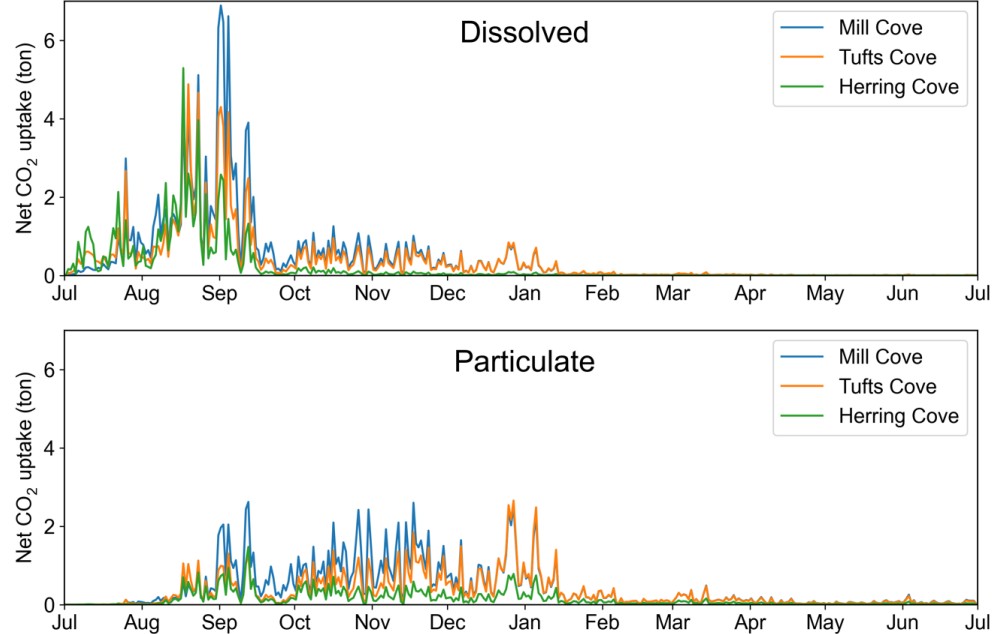


Figure 8. Time series of spatially integrated net $CO_2$ uptake in the simulations with dissolved (top) and particulate (bottom) feedstock.



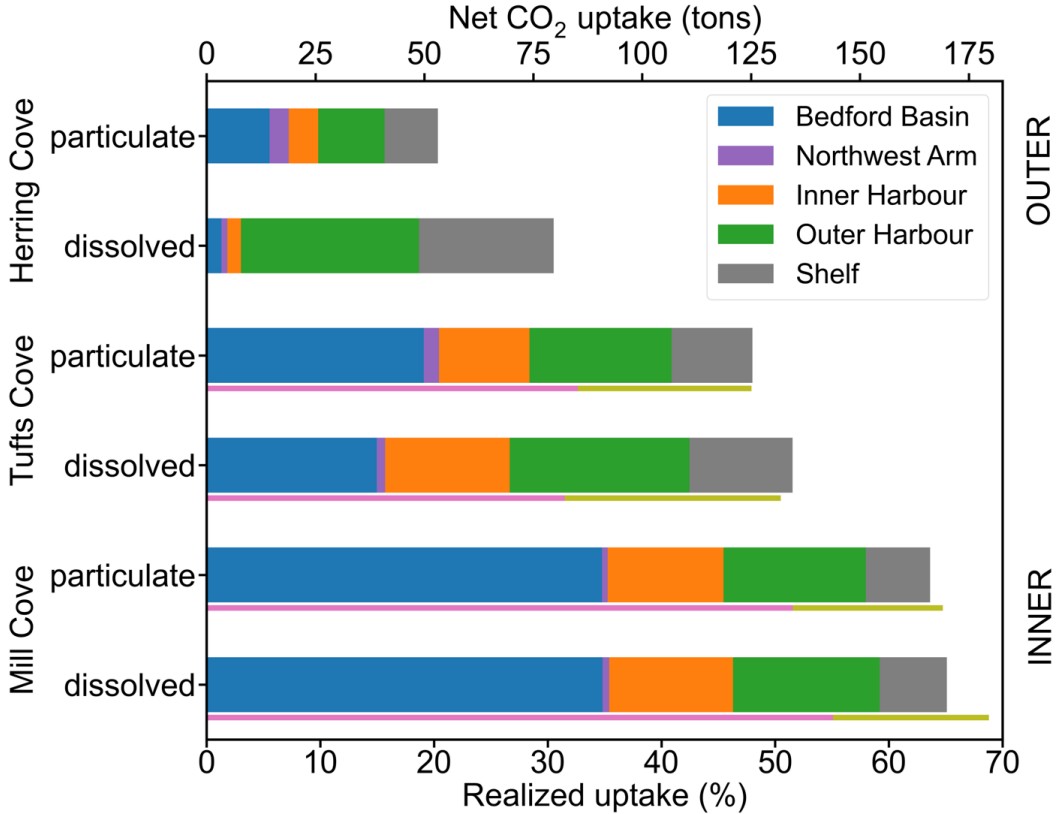

Figure 9. Realized net $CO_2$ uptake for each simulation. The colors in the legend indicate the total uptake in each area of the model grid. Pink and olive-green lines under the Tufts Cove – dissolved bar indicate the realized uptake in ROMS-H3 (inside harbour) and in the outer area of ROMS-H2 (outside harbour), respectively (see Figure 1d). Maximum achievable net $CO_2$ uptake from the alkalinity addition was estimated to be 261 tons using a theoretical uptake efficiency of 1.34 g $CO_2$ per g $Mg(OH)_2$ (see Section 5.2.1). The realized uptake is the relative difference between maximum achievable net uptake and simulated net uptake.



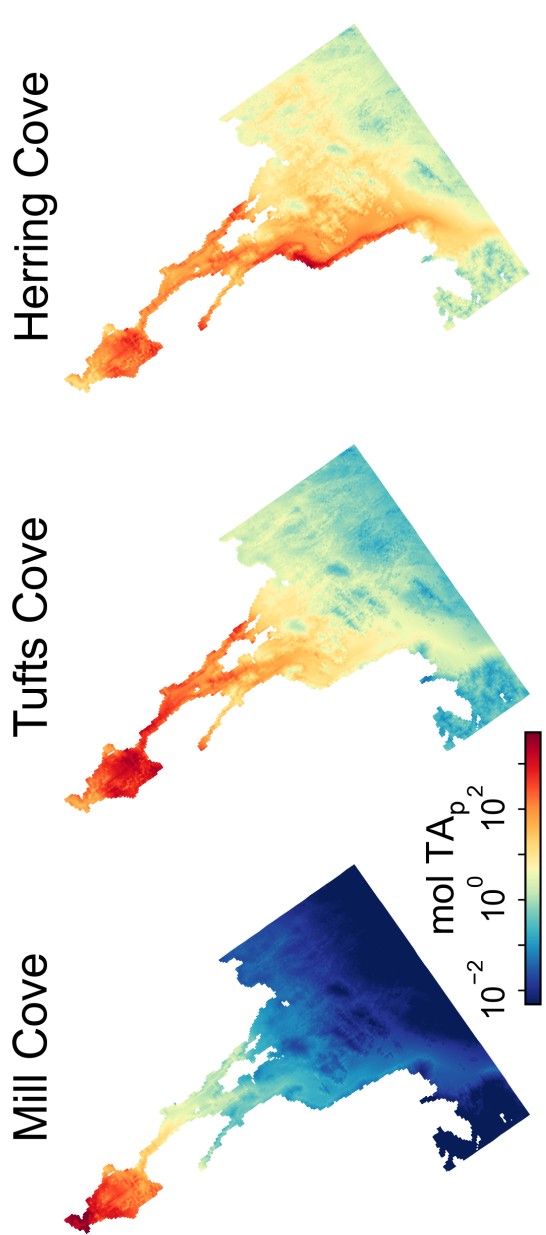

965        Figure 10. Average distribution of particulate feedstock during the simulations with addition at the three locations.



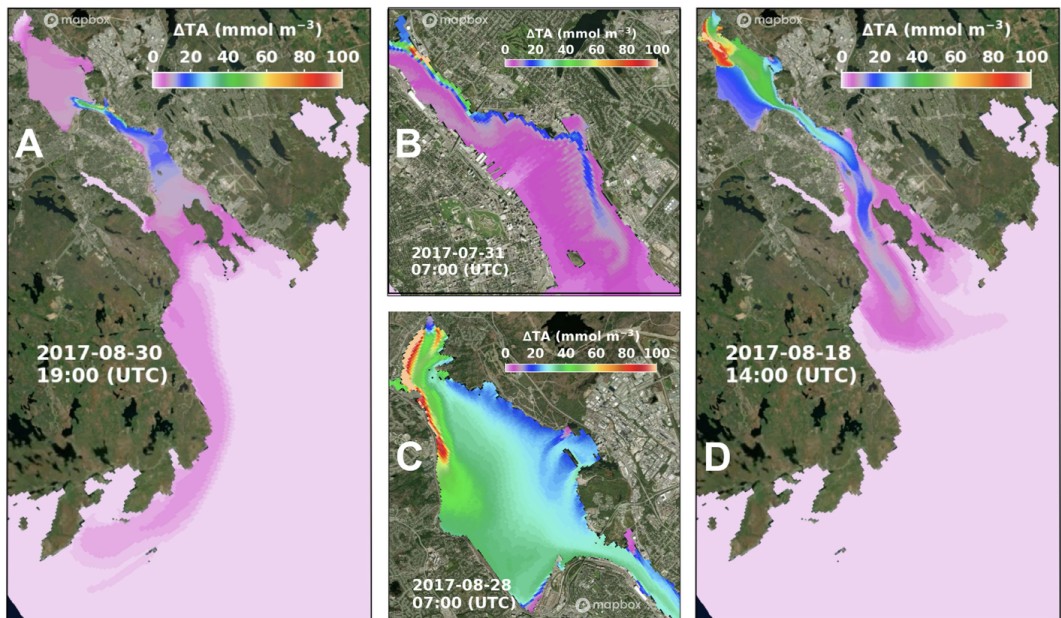

Figure 11. Snapshots of surface ΔTA in the nested simulations with ROMS-H2 and ROMS-H3 for a release of fully dissolved feedstock at Tufts Cove (a,b) and at Mill Cove (c,d). Panels b and c show a zoom in on the inner Harbour (b) and the Bedford Basin (c). Satellite images were generated with MapBox © Mapbox
