# Peer review of "A high-resolution nested model to study the effects of alkalinity additions in Halifax Harbour, a mid-latitude coastal fjord"

_EGUsphere, 2025_

## Author Comment (AC1)

**Detailed responses to reviewer 1** (reviewer comments are included in black, responses in blue font)

**General comments**

The authors present a nested regional ocean model of Halifax Harbour and part of the Scotian shelf which is validated against measurements. A simple dissolution model is implemented and pulse releases of an alkaline effluent are modelled, consisting of a mix of dissolved and particulate alkalinity. The subsequent changes in alkalinity and DIC (from the induced CO2 uptake) are evaluated and analyzed.

Overall the manuscript is well laid out, focused and easy to follow. The simulations presented establish an important standard of rigor for future OAE deployments in other areas. I recommend publication.

**Response:** We appreciate the positive assessment and the constructive feedback. We have addressed all the comments as described in the detailed responses below.

**Specific comments**

**Comment:**

1. The authors show that alkalinity addition inside a natural enclosed harbour enables a substantial fraction of the theoretically maximal  $CO_2$  uptake to occur quickly and within the simulation domain, due to the long residence time and relatively shallow waters. As pointed out in L556-559, this makes MRV much easier both experimentally and from a simulation perspective. Of course the flipside of this is that a confined body of water which does not quickly spread any added  $\Delta TA$  over large ocean areas will also limit the total sustained alkalinity addition rate in that area, limiting scaling of OAE.

It would be useful to add an estimation of this in the manuscript. For a rough, first pass estimate, perhaps one could assume that the response of  $\Delta TA$  and  $\Delta DIC$  are roughly additive and linear with respect to addition rate. Then, for each of the three locations, one could calculate what the maximum addition rate would be which would raise the maximal  $\Delta pH$  to some acceptable limit (what that limit is is of course arbitrary, but perhaps something conservative like +0.1 or +0.05 units would be illustrative).

Another approach would be perhaps to examine the export rate of alkalinity out of the simulation boundary and try to estimate what sustained alkalinity addition rate (rather than a pulse) could be achieved, again within some  $\Delta pH$  or  $\Delta TA$  limit set within the domain.

A discussion of this and the trade-offs of release locations would be useful to the reader to understand better what sort of scale OAE can achieve.

**Response:** These are interesting and highly relevant comments. Regarding the trade-off between measurability of the alkalinity signal in the harbour (which is helped by the high residence time of the system) and the risk of breaching regulatory and environmental

thresholds (which is elevated in this system because of the long residence time) we would like to refer to Wang et al. (2025) where this was investigated using an "exposure" metric and many simulations with release from Tufts Cove and Mill Cove of different feedstocks at various dosing rates. Wang et al (2025) used a similar physical setup with a particle dissolution model. They found that exposure rate is much higher when feedstock is released from Mill Cove, especially in summer (higher residence time) for slow dissolving/fast sinking particulate feedstock. In their experiments the addition of slow sinking, slow dissolving particles at Tufts Cove resulted in lowest exposure.

We did discuss Wang et al. (2025) results regarding exposure risks in Section 6.3 but will expand this discussion in the revised manuscript.

Regarding the implications for scalability, this is the subject of a comprehensive analysis in a forthcoming manuscript.

**Comment:**

The treatment of dissolution as an exponential decay process (i.e. dTAp/dt = -k TAp) was surprising at first glance. Usually dissolution of particular matter is treated with a shrinking core model, where the dissolution rate has units of mol cm-2 s-1, the radius of particles shrinks linearly and fully dissolves in a finite amount of time. For a very narrow (as indicated in L335, "a particle size of  $12\mu m$ ") or uniform distribution of particle sizes I believe an exponential dissolution curve is only a mediocre fit.

I can see that an exponential model could perhaps capture the behaviour of a gaussian or log-normal distribution of particle sizes, but a short discussion of this and a justification of the choice of model here would be helpful.

**Response:** In Section 5.1 of the revised manuscript, we will discuss the choice of an exponential decay for the dissolution of particles and in Section 5.2 we will provide more details on the choice of the dissolution parameter in the experiments with particulate feedstocks.

**Comment:**

3. L317
$$w_{p}TA_{p}$$
 term:

It's unclear to me how the sinking term is applied. As written it looks like there is an exponential decay, i.e. each time step some fraction of TA\_p is lost to sinking from any given simulation grid voxel. What happens to that TA\_p? Does it get added to the cell below, until the bottom cell is reached after which it disappears in to the sediment? Or does the model assume the sunk particles are removed completely (i.e. they sink out entirely at a rate of W\_p\*TA\_p from anywhere in the column?). As currently written it seems more like it's the latter, as there is no term that accounts for sinking particles that arrive from a cell above (i was expecting a second term like  $+w_p*TA_p^{z=i-1}$ )

Please clarify how the sinking mechanism is implemented and justify its construction.

The sinking rate is stated as 5.5 m $^{-1}$  later (L337) but that can't be w\_p since the units wouldn't be right (w\_p should have units of inverse time, like k\_{diss}). How is w\_p calculated from the 5.5m $^{-1}$ ?

**Response:** We will clarify this part in the revised manuscript. The confusion originates from a typo in the units of  $w_p$ , which should have been m s-1, and the formulation of sinking in Eq. 8, which should have been  $w_p \frac{\partial TA_p}{\partial z}$ .

Sinking occurs between the vertical layers of the model. As the reviewer mentioned, sinking is a source term to the layer below and a loss term from the layer above (except for the surface layer). In the bottom layer, particulate material accumulates and a fraction is lost through the incorporation into the sediment (Eq. 14, see response to comment below).

**Comment:**

4. L326 The treatment of sediment loss in layer N is a little unclear. It says a term is "added" to  $\partial \Delta TA/\partial t$ ? Or does this replace the regular dissolution term in  $\partial \Delta TA/\partial t$  (last term in Equation 9)? It might be clearer here to just rewrite the full Equation 9 (and perhaps Equation 8) in the case of the bottom cell, for clarity.

It's also confusing to me that the loss of TAp due to sinking/burial is already explicitly treated in equation 8 using w\_p and then it's treated again here with the \theta\_{loss} term. Is \theta\_{loss} a constant? Or is it calculated from w\_p?

**Response:** We will update Eqs 8 and 9 for clarity. Sinking and burial are two different processes (see also response above). In the bottom layer, sinking material is immediately resuspended and therefore accumulates. Instead of a sinking loss term there, we assume that a constant fraction of TAp present in the bottom layer and thus in contact with the sediment is incorporated into the sediment through bioturbation and thus lost from the system.

**Comment:**

5. L424 The comparison of H2 and H3 is very interesting and suggests perhaps a resolution as high as H3 isn't necessary. A similar comparison of H1 vs H2 would also be useful if the releases can be reasonably implemented at the coarsest level. Even if the release location would have to be assumed to be wider or poorly matched in terms of exact location, injection of the same amount of alkalinity in the coarsest model could be interesting to determine to what extent the H2 level is required.

**Response:** See also response to Comment 3 by Reviewer 3. H1 was designed to provide reasonably accurate boundary conditions to H2 on the Scotian Shelf but remains quite coarse (~760 m) with respect to the Halifax Harbour. For comparison, H2 has a

resolution of 150 m. The shape and circulation of the Harbour is not at all well resolved in H1, which affects residence time. We note that the purpose of this model is to accurately represent the dynamics in Halifax Harbour and the purpose of the manuscript is to describe this.

The Reviewer's suggestion to compare at the coarser resolutions is a good one for a study that is aimed at assessing how resolution affects transport and dispersion of alkalinity and DIC on the shelf. However, we wouldn't use H1 for this purpose because it only covers a small portion of the shelf (note that alkalinity, once it leaves Halifax Harbour, is transported out of the H2 domain in just 10 to 14 days). We are working with a larger-scale ROMS model, described in Ohashi et al. (2024, https://gmd.copernicus.org/articles/17/8697/2024/), that has a resolution similar to H1 and intend to compare this with coarser-resolution models.

**Comment:**

6. L769 It was a surprise to read here that the sediment loss term was set to zero. I feel like this should have been mentioned earlier, perhaps even right when the loss term(s) are introduced in L317ff. Is both wp and \theta\_{loss} set to zero or just the latter? If it's just the latter, does the model currently just settle all the particles on the floor and let them dissolve from there until completely dissolved?

**Response:** For clarity, the loss term set to zero will be mentioned in Section 5.1. in the revised manuscript.

For an explanation on sinking and sediment loss, see response to Comment 3 above. Indeed, all particles settle to the bottom in the current setup and dissolve. Particles are also transported by current and can be resuspended by vertical mixing. We chose to set sediment loss to zero because unfortunately we do not have information on the influence of the sediment on Brucite particles in the Halifax Harbour, as mentioned in L569.

**Comment:**

7. L120: I assume the conversion factor is 1025 kg m^-3, not 1.025kg m^-3 (remove dot or change dot to comma)

**Response:** We did assume a water density of  $1025 \text{ kg m}^{-3}$  but the conversion factor is 1.025 because we convert  $\mu$ mol to mmol. We will clarify this conversion in the revised manuscript.

**Comment:**

8. L243 In equation (3), it appears that the parameter "c1" is duplicate as a coefficient to t and as an exponent. Likely it is meant to be c2 instead?

**Response:** Indeed, this is a typo, it should be c2.

**Specific comments**

**Comment:**

9. L325 change to "is added that mimics" or "is added to mimic"

Response: Done.

**Comment:**

10. L331 "1.29 ml s-1", exponentiate the "-1"

Response: Done.

**Comment:**

11. L475 In such cases,

Response: Done.

**Comment:**

12. Fig.1D consider using a different color scheme for the bathymetry as the scale is different.

**Response:** We feel that an alternate color scheme is not necessary since the color bar is available in Figures 1c and 1d. However, since the color bar is missing from Figure 1a (same as in Figure 1c) we will add it in the revised manuscript.

**Comment:**

13. Figs. 3, 5,6,7, 10: Is it possible to indicate the release location in these plots with a small black arrow or similar. I know they are shown in Fig 1 D, but it would be very helpful to have that info on each of the other plots too.

**Response:** Yes, we will add a dot showing the release location in these plots.

**Comment:**

14. Figure 7: It would be nice to add a horizontal dashed line to the two graphs indicating the theoretical maximum uptake (at your CO2 efficiency of 0.89) to get a sense for what fraction of the ultimate uptake occurs within the simulation domains.

**Response:** Yes, we will add an additional y-axis on the right indicating the realized uptake (this information is also available in Figure 9).

**Comment:**

15. Fig S4-S8 The observations of the depth profiles are sparse enough in time that it's difficult to assess visually how closely the corresponding model predictions match. Perhaps, for each observation time and depth simply make a scatter plot against the corresponding prediction value? Could be color coded by depth perhaps to see if correlation is better at surface vs depth.

**Response:** This information is somewhat already synthetized in the statistics (Table 1) but for a visual comparison we will also add 1:1 plots in the supporting material of the revised manuscript.

---

## Author Comment (AC2)

**Detailed responses to reviewer 2** (reviewer comments are included in black, responses in blue font)

**General comments**

This is a very interesting and thoughtful paper on a numerical modelling approach to detect and evaluate the effects of OAE. It represents a significant step forward towards realistic simulations of an actual alkalinity release field experiment. I think the paper can be accepted for publication after moderate revisions and clarifications.

There are several questions I hope the authors can clarify.

**Response:** We appreciate the constructive comments. We will revise the manuscript accordingly and provide point-by-point responses to the reviewer's comments below.

**Specific Comments/Questions**

**Comment:**

1. I like the approach to separate the slurry additions into dissolved TA input and particulate form which later dissolve and sink. This is more realistic than the previous modelling approach which adds TA in dissolved form. It also counts for lost TA due to particle sinking onto the seabed. It was interesting to see the result that the maximum CO2 uptake from this mixture is lower. On the other hand, the more realistic model representation comes at the expense of introducing three additional parameters: the particle dissolution rate, the particle sinking rate and the fraction of slurry particles incorporated onto the sediment, the last of which would be difficult to estimate. In reality, there would be a size spectrum of slurry particles which dissolve and sink. As shown in Fig. S10 in Wang et al. (2025), the sinking velocity varies by two orders of magnitude for alkaline feedstocks of various sizes. It is possible that the dissolution rate may also change with the particle size. Wang et al. (2025) also showed the results are very sensitive to the particle dissolution rate and sink velocity. How does one choose one "representative" particle with a particular size, dissolution rate and sinking velocity? I understand the need to keep the model manageable, but these are model assumptions that could be discussed. There are approaches to model a size spectrum of bubbles generated by breaking waves (Garrett et al., 2000). The bubbles are injected into the upper ocean, rise due to buoyancy and dissolve under partial pressure differences. Maybe some of these modelling approaches could be discussed. Sediment transport modelling has to deal with a spectrum of particle sizes too and it is well known the settling site of sediment depends critically on the particle size.

**Response:** Given the high spatial model resolution, simulating a particle size spectrum is not feasible. It would be computationally too expensive. Detailed models of size-spectra, like the one for bubbles and those used for sediment transport either use simplified physical models (e.g. 1-dimensional) or are run only for a few weeks (in the case of sediment transport coupled to a 3-dimensional ROMS model). Here we used a highly resolved physical model of the Harbour and had to compromise on using a single parameter for dissolution and sinking rates to represent the bulk of particle sizes. This is a

limitation of the model. To clarify this point, we will add text in Section 5.2 to justify our choice. We will also mention how the dissolution and sinking parameters used for the experiments with particulate feedstock were calculated. Finally, we will discuss this limitation in Section 6.4 (Current limitations and future development).

**Comment:**

2. I am also curious about the author's approach to use a high-resolution hydrodynamic model but a simplified biogeochemical model. I can understand the need for high resolution hydrodynamic to resolve the near-field transport and dispersion of added slurries in the inner model domains but do not quite understand the use of a simplified biogeochemical model. Was it due to the high computational cost of the full biogeochemical model? I thought the biogeochemical model can be run very efficiently if done on an offline mode. It would be good to discuss why the authors took this modelling approach. It will be instructive to other modelers.

**Response:** As mentioned in Section 4, the reduced complexity biogeochemical model was designed to improve computational efficiency. In our experience, offline models do not match well the more accurate results from online simulations which is why we don't use them. The simplified biogeochemical model has an intermediate level of complexity that we deem to be fit-for-purpose. If we were to run the full biogeochemical model, there would be no direct interaction between the nitrogen and phosphorus species in that model and the carbonate system. The simplified model delivers what we need in terms of describing the seasonal cycle of background DIC. The comparison with observations at the monitoring station in the Bedford Basin shows that despite the simplification the model can simulate the background carbonate system appropriately. Further complexity is therefore not needed in the context of OAE. We will expand the justification for using a reduced complexity biogeochemical model in Section 4 of the revised manuscript.

**Comment:**

3. There is also this broad question how we can validate the model results and document the OAE effects. The authors did a lot of model validation without OAE but none for the model results with OAE. How can the model help the documentation and verification of alkalinity addition? The latter is a nagging issue facing all OAE studies, due to a combination of large natural variability in the carbonate system and the policy restriction/regulation on exposure impacts.

**Response:** The manuscript describes the OAE model and provides insight on the effects of release locations and feedstock type in the Halifax Harbour. We did not intent to carry out MRV with these experiments, but the reviewer makes a good point about the need of validating the model with OAE in the case of MRV. This will be the focus of a follow-up study, as mentioned in the Conclusions (L593). In this context, a set of observations needs to be collected directly at and near the dosing location.

---

## Author Comment (AC3)

**Detailed responses to reviewer 3** (reviewer comments are included in black, responses in blue font)

**Overview**

The manuscript describes a novel BGC model that is run within a physical model (ROMS) which itself can be run at 3 described resolutions (ROMS-H1, ROMS-H2, ROMS-H3). ROMS-H2 is the workhorse resolution here. The novel BGC model here is a cut-down version of a more complete BGC model (not clearly named here) that essentially simplifies the BGC to DIC, TA and O2, with major missing processes parameterised and described here. This model is then used in a series of OAE experiments. The manuscript details coupled physics-biogeochemistry model specifically designed for OAE in a nested grid configuration and reduced biogeochemistry with increasing spatial resolution from Scotian Shelf to Halifax harbour. To ensure that the model is suitable for the location, hindcast simulation of the model is validated against observation on the shelf. Alkalinity enhancement experiment is simulated at the inner, mid, and outer harbour with two different feedstocks; fully dissolved and fully particulate. Then the effect of alkalinity addition to the carbonate system is analysed. A major conclusion is around the relative success (69%) of CO2 absorption driven by OAE within the modelled coastal domain.

Overall, the manuscript is an interesting investigation into OAE in a very specific locale, Halifax Harbour. In most places, the manuscript is written quite well, and the model has been generally shown to capture the observations. The manuscript also explores different locations of alkalinity addition and types of feedstocks, which are relevant and can be insightful for field study and MRV. However, while we generally appreciated it, there are a few suggestions that would make the manuscript stronger.

**Response:** We appreciate the assessment of the manuscript and the feedback. We provide point-by-point responses to the reviewer's comments below

**General comments**

**Comment:**

1. A general comment we'd make is that the structure of the manuscript impedes its interpretation. In particular, it merges model description, experiment design and results from the OAE side of the work into a single and lengthy section. There's nothing special in the work that precludes a conventional method-results-discussion structure, so please reformulate the manuscript this way.

**Response:** The structure of the manuscript is probably a matter of personal preference. Reviewer 1 said that the manuscript was "well laid out, focused and easy to follow". We chose to split the model description/results into three parts (physics-shelf BGC, reduced BGC, OAE module) for clarity. Using a more conventional structure would not improve the interpretation because the three parts mentioned above would be mixed, likely introducing confusion for the reader. We feel that the current structure of the manuscript is best to describe the model and present the results of the simulations with OAE.

**Comment:**

2. Another general comment is that the use of a novel BGC model here introduces the requirement for a lengthy digression into the formulation and skill of this model. Ideally, such a model would be described in a separate manuscript and then used for the problem at hand. As written, it is sometimes unclear whether the paper's primary aim is to describe the novel OBGC model itself or to address OAE research problem, which in our opinion, makes the narrative harder to follow. We would suggest clarifying the manuscript's focus and streamlining the model description, so that this description is largely moved to supplementary and shortened in the main body.

**Response:** We disagree that the introduction of the model is a lengthy digression. In fact, the description of this model is one of the main points of this paper (i.e. this is the "separate manuscript" describing the model that the reviewers seem to be asking for). The manuscript investigates alkalinity additions in an idealized setup (Section 5). An application of the model to the ongoing OAE field studies in Halifax Harbour will be a separate manuscript. Given the structure of the manuscript, a reader who is not interested in methodological details, can focus on the OAE experiments. We moved most of the validation and additional figures and tables to the supplement to keep the main body clear. We also moved some of the general BGC equations to the supplement. We hope that the manuscript is a good compromise between methodological details and readability for most readers.

**Comment:**

3. On a related point, because the model used here is novel, it would be extremely beneficial to compare it with its parent model. At present, the validation in Supplementary Material appears to compare / validate the parent and novel models separately rather than together. For example, Figures S3 to S8 appear to use the parent, while Figures S10-S19 appear to use the novel model, but there are none intercomparing the shared properties of the models with observations. Given how different the models appear to be in their state variables and formulation, it is critical for readers to understand how strong the relationship between the models is.

**Response:** See also response to Comment 5 by Reviewer 1. ROMS-H1 simulates circulation and biogeochemistry on the shelf, whereas ROMS-H2 and ROMS-H3 simulate background carbonates and OAE in the nearshore and inside the Halifax Harbour. While the later can be compared (e.g., Section 5.2.4, see also Wang et al., 2025), the purpose of ROMS-H1 is to provide accurate boundary conditions to ROMS-H2, which then simulates the dynamics of the Halifax Harbour. We do not feel that comparing ROMS-H1 and ROMS-H2 at the same stations (ST2 and BBMP) would provide much insight since ST2 is located near the open boundary of ROMS-H2 and ROMS-H1 does not resolve the Halifax Harbour circulation well (BBMP).

**Comment:**

4. We would also suggest that the manuscript would be more complete if the authors also discussed other diagnostics that are relevant for impact assessments. Such as how OAE affects Halifax Harbour's pH and saturation states (e.g.  $\Omega$  aragonite) over the course of alkalinity addition. Those are liable to be important for natural ecosystems in the region.

**Response:** pH and saturation states are important variables in the context of OAE, as an indicator for stress but also because of the regulatory limit on pH and the potential for precipitation with higher saturation states. Looking at the effect of alkalinity additions on these variables is important and will be the focus of a follow-up manuscript.

**Specific comments**

**Comment:**

5. Abstract: A little light on actual results. I would have expected a statement regarding either the efficacy or challenges encountered during the work.

**Response:** We will add a statement about the outcome of the OAE simulations at the end of the abstract. Further, we note that one of the purposes of this manuscript is to describe the model (see our response to comment 2).

**Comment:**

6. Line 90: some models, e.g. Palmieri & Yool (2024), assumes that particulate material is added, and uses the calculated dissolution rate of this to specify the flux. In this specific model, yes, the model only sees a flux of TA, but this is achieved through the temperature-dependent dissolution rate of an implicit particulate source (which is assumed to have already settled on the seafloor).

**Response:** The sentenced will be modified as follows:

"Most models assume the addition of fully dissolved rather than particulate material (e.g., Kwiatkowski et al., 2023; Nagwekar et al., 2024; Zhou et al., 2025), whereas mineral feedstock particles will dissolve over time (Schulz et al., 2023). Palmiéri and Yool (2024) assumed dissolution but of an inert particulate feedstock at the sediment-water interface."

**Comment:**

7. Line 96-103: It might have expected to see some articulation of the key research questions here. This seems primarily a breakdown of how the manuscript is organised. (Of which, see below.)

**Response:** This paragraph will be modified as follows:

"To fill the gaps described above, we present a high-resolution, coupled physical-biogeochemical-addition-dissolution model that is designed to support OAE research in coastal environments. The model builds on Wang et al. (2025) to include oxygen and carbonate system chemistry in a realistic alkalinity addition setting. The model is validated and tested for Halifax Harbour. The key question that is investigated with this novel model is how feedstock type and dosing location influence alkalinity dispersion and net CO2 air-sea flux in the Halifax Harbour and surrounding nearshore waters.

The observations used for the study are described in Section 2. The circulation model is presented in Section 3 and the results of an eight-year simulation (2016-2023) validated against observations on the shelf and in the harbour are presented. The biogeochemical

model is described in Section 4 and validated for 2016-2023. The addition model is then presented and a series of OAE simulations carried out in Section 5. The overall results are discussed in Section 6."

**Comment:**

8. Methods: There are 6.5 pages of model description before we get to the OAE part. Ideally, the model used would be an existing model, previously described elsewhere. However, we are where we are. We would suggest abbreviating this to a ~1 page summary that cites prior work and moving the more expansive text to an appendix. It's useful – in fact, given this is a novel model, \*critical\* – to have all of this, but this level of detail tends to distract from the focus of the paper.

**Response:** Again, when the reviewers say here that we should have used an "existing model" that should have been "previously described elsewhere" they are missing the point that this is the first time it is described. It will be used for more in-depth studies in forthcoming manuscripts. The information included in the main manuscript only covers the novel parts of the model and we feel this is critical to the understanding of the paper. The validation figures that are not essential were included to the supporting material. Also see response to Comment 2 above.

**Comment:**

9. Ln. 137: It would be helpful if the BGC models here were given names so that they can be clearly identified, and clearly separated from the physical frameworks they are coupled to.

**Response:** The full biogeochemical model used on the shelf (ROMS-H1 grid) is referred as "the explicit biogeochemical model" throughout the manuscript, whereas the model developed for alkalinity addition and used in H2 and H3 grids was named "the reduced complexity biogeochemical model". Given the structure of the manuscript the explicit biological model is first discussed in Section 3 and then the reduced complexity biogeochemical model is discussed in Section 4. We feel that this should avoid confusion. For clarity, we add the following in Section 3 (L150):

"ROMS-H2 and ROMS-H3 are coupled with the reduced biogeochemical model (see Section 4)."

**Comment:**

10. Line 300: Section 5 appears to combine model description with model results. This doesn't seem a helpful way to present the work. We would suggest creating / merging into separate sections to 1. describe the model (part of methods), and 2. describe findings from its use (the results section). Similarly, the validation of the model ahead of its use could either be arranged formally part of the methods (as it kind-of is now) or moved to be part of the results.

**Response:** See response to Comment 1 above. We feel that merging Section 5 with the previous sections would lead to confusion. The current structure where all alkalinity

addition (model description, experiments, results) is in the same section seems easier to follow.

**Comment:**

11. Line 301-303: a verbal overview description of the OAE scheme might fit well here. Specifically, that OAE TA is added to the ocean in dissolved TA and particulate TA forms (the latter requiring a new tracer), and that the latter dissolves into the former with time. Also, this whole model description completely overlooks Figure 2 which rather clearly indicates how the model works.

**Response:** The introductory paragraph will be modified as follows:

"To simulate alkaline feedstock addition (often in a liquid form, e.g. slurry) and the effect of added alkalinity on the carbonate system, several tracers were added to the biogeochemical model. An overview of this model, including the addition module, is presented in Fig. 2. The alkalinity addition module is described in detail below. It was designed so that a single simulation can be used to calculate the difference between the realistic addition case and counterfactual"

**Comment:**

12. Line 306: It could be helpful to explain why new delta tracers were added rather than the more conventional compare-and-contrast with a control simulation. One can imagine what the explanation is, but readers would benefit from understanding this.

**Response:** Models running on HPC systems are typically not bit-reproducible making the conventional approach to compare-and-contrast with a control simulation not ideal. Additionally, running simulations is essential to MRV but has also a carbon footprint (energy consumption), which is improved with the use of dual tracers.

**Comment:**

13. Ln. 310-312: This describes the split between dissolved and particulate TA additions but does not clarify what the fraction is or how it may vary. It also seems like this is something that isn't explored elsewhere in the manuscript, e.g. a sensitivity analysis on the importance of this fraction might be expected. In any case, one would at least expect to be told what its value was, or – if this is variable – under what conditions it varies.

**Response:** The value of  $\theta$  depends on the characteristics of the feedstock. Since we used either fully dissolved or fully particulate feedstock in the experiments, we only set  $\theta$  to 0 or 1. Therefore, our results already show the maximum range within which sensitivity experiments for  $\theta$  would lie.

In the revised manuscript the sentence will be clarified as follows:

"The allocation of  $TA_{in}$  into the dissolved ( $\Delta TA$ ) and particulate ( $TA_P$ ) pools is set by the parameter  $\theta_{P:D}$ . This representation of dissolution of the particulate stock builds on the formulation in Wang et al. (2025). The value of  $\theta_{P:D}$  varies for each feedstock and needs to be assessed prior to dosing. Simulation results (see below) from experiments with fully dissolved ( $\theta_{P:D} = 0$ ) or fully particulate ( $\theta_{P:D} = 1$ ) indicate the maximum effect of  $\theta_{P:D}$ ."

The sentence introducing the experiments (L335) will be changed to:

"... each with two different feedstocks, either fully dissolved ( $\theta_{P:D} = 0$ ) or fully particulate ( $\theta_{P:D} = 1$ )."

**Comment:**

14. Eqn. 8-9: We would suggest adding the equation for TA so that the relationships between TAp and delta-TA are easy to understand. Same for DIC.

**Response:** The equations for TA and DIC are available in the supplementary material (mentioned on L229). Here we focus on the alkalinity addition module.  $\Delta$ TA and  $\Delta$ DIC are independent variables from TA and DIC. They are only used in combination to calculate air-sea CO2 flux.

**Comment:**

15. Ln. 329: This section is sorely missing a clear non-narrative description of the experiments undertaken and the simulations performed. A table listing the simulations and their roles would make it very easy for readers to understand the work undertaken.

**Response:** This table will be added to the revised manuscript.

**Comment:**

16. Ln. 329: Technically, it could be argue that this subsection is methods, but the organisation of subsection 5.2 is compromised by a subsubsection 5.2.1 which is more results than methods. Reorganising into the conventional methods-results-discussion format would greatly improve this manuscript.

**Response:** See response to Comment 1 above.

**Comment:**

17. Ln. 331: superscript typo, 1.29 mol s-1

Response: Done.

**Comment:**

18. Ln. 363-371 and Ln. 383-403: these sections of text quantify the results found verbally rather than, more obviously, in a table (or tables). This would allow the reader to clearly understand distinctions being drawn between experiments and locations that currently require the reader to hold a lot in their heads at once just to make basic sense of the results.

**Response:** The information discussed in these sections is shown in Figures 7 and 9, which is why a table was not included. The information will be summarized in a new table in the revised manuscript.

**Comment:**

19. Ln. 364: The manuscript presents a maximum achievable uptake of 0.89 mol CO2 per mol TA, perhaps we are missing something, but can the authors elaborate more on how

this number came up? Will this number change with seawater condition / location / model resolution?

**Response:** This number was calculated based on simulated conditions in the Halifax Harbour, as mentioned L365. We will provide more details on this calculation in the revised manuscript.

**Comment:**

20. Ln. 448: The balance in this manuscript between the base model and the OAE experiments is generally off, and this section (6.1) continues this. Manuscripts like this one generally present just enough information about the base model to satisfy readers that it's an appropriate choice for the experiments in question. Also, and more importantly, it blends in details about the OAE experiments that might better be discussed in the relevant section (6.2, 6.3).

**Response:** See response to Comment 1 above. Model validation is important, and Section 6.1 discusses both the validation of the current model and the possibility of validating a relocated version of the model. We feel that it is better done in a separate section than blended into Sections 2 and 3.

**Comment:**

21. Ln. 449-450: Why "not surprisingly"? Also, similar performance to what? Is there maybe a grammatical error in this sentence.

**Response:** For clarity, we will modify the sentence as follows:

"The circulation model is modified from the nested model of Wang et al. (2025) using rotated and refined grids. Not surprisingly, the physical model had similar performance to Wang et al.'s version when compared with observations (Table 1)."

**Comment:**

22. Ln. 551: "Tufts Cove is the location for alkalinity ..." should this be "Tufts Cove is the *ideal* location ..."?

**Response:** The Halifax Harbour is a site for alkalinity addition since 2023. Dosing is done at Tufts Cove. Our results did not show that Tufts Cove is an ideal location for dosing.

**Comment:**

23. Ln. 591: "These results stress the importance of operational design as well as the use of high-resolution regional models when quantifying additionality." – Can you be clearer on what, specifically, is meant by "operational design" here? Is it geographical / depth choice of TA release, the balance of dissolved / slurry supply of TA, the timing of application of slurry (w.r.t. tides)?

**Response:** The sentence will be changed to:

"These results stress the importance of operational design, such as dosing location, type (dissolved, particulate), slurry properties (dissolution rate, sinking speed), as well as the use of high-resolution regional models when quantifying additionality."

**Comment:**

24. Figure 2: This shows pH in a similar way to model state variables. But it's a calculated property rather than a state variable. Maybe alter the line style of the box so that it's clear that it's not a state variable?

**Response:** The line style will be changed to dashed.

**Comment:**

25. Figure 3: while I have used Ocean Data View-style palettes like this before in my own work, I now appreciate that they present particular difficulties for readers with colour vision issues. To which end, please convert this figure to a more suitable palette, plus subsequent ones using the same palette.

**Response:** We will change to a colorblind friendly palette.

**Comment:**

26. Figure 5-6: This seems to use a different geographical domain to Figure 3 onward. Is there a reason for this? More generally, why do these figures include satellite imagery while the others don't? (Figure 1d also shows slightly different domain limits than in Figure 3.)

**Response:** The limits of the three model domains are presented in Figure 1c. However, it is not always pertinent to show the entire model domain. For example, enlarging Figures 3 and 7 would only show additional values ~0 (pink) but the relevant areas (Halifax Harbour) would be smaller. This is mentioned in the captions. Showing the entire grid area in the other plots would not add more information. For clarity the following will be added to the captions: "(offshore areas were excluded from the map)".

We did not include satellite imagery on Figures 3, 7 and 10 because these figures are compact and satellite imagery did not work well in this context.

**Comment:**

27. Figure 7: needs rotating.

**Response:** The figure was rotated to be larger for the review but will not be rotated for final submission.

**Comment:**

28. Figure 10: needs rotating.

**Response:** See response to Comment 27 above.